# Relationship Between Voice Analysis and Functional Status in Patients with Amyotrophic Lateral Sclerosis

**DOI:** 10.3390/audiolres15030053

**Published:** 2025-05-07

**Authors:** Margarita Pérez-Bonilla, Paola Díaz Borrego, Marina Mora-Ortiz, Roberto Fernández-Baillo, María Nieves Muñoz-Alcaraz, Fernando J. Mayordomo-Riera, Eloy Girela López

**Affiliations:** 1Physical Medicine & Rehabilitation, Reina Sofia University Hospital, 14004 Córdoba, Spain; sr1maif@uco.es; 2Department of the Applied Physics, Radiology and Physical Medicine, Faculty of Medicine and Nursing of Córdoba, 14004 Córdoba, Spain; 3Physical Medicine & Rehabilitation, Virgen Macarena University Hospital, 41009 Seville, Spain; paola.diaz.sspa@juntadeandalucia.es; 4Maimonides Biomedical Research Institute of Córdoba (IMIBIC), Reina Sofía University Hospital, University of Córdoba, 14004 Córdoba, Spain; marina.mora@imibic.org; 5Department of Human Anatomy and Embryology, Faculty of Medicine, University of Alcalá, 28801 Madrid, Spain; roberto.fernandezbai@uah.es; 6Department of Morphological and Sociosanitary Sciences, Faculty of Medicine and Nursing of Córdoba, 14004 Córdoba, Spain; ft1gilpe@uco.es

**Keywords:** Amyotrophic Lateral Sclerosis, acoustic analysis, dysarthria, functionality

## Abstract

**Background:** Amyotrophic Lateral Sclerosis (ALS) is a progressive neurodegenerative disease affecting both upper and lower motor neurons, with bulbar dysfunction manifesting in up to 80% of patients. Dysarthria, characterized by impaired speech production, is common in ALS and often correlates with disease severity. Voice analysis has emerged as a promising tool for detecting disease progression and monitoring functional status. **Methods:** This study investigates acoustic and biomechanical voice alterations in ALS patients and their association with clinical measures of functional independence. A descriptive observational case series study was conducted, involving 43 ALS patients and 43 age and sex matched controls with non-neurological voice disorders. Sustained vowel /a/ recordings were obtained and analyzed using Voice Clinical Systems^®^ and Praat software (version 6.2.22). Biomechanical and acoustic parameters were correlated with ALS Functional Rating Scale-Revised (ALSFRS-R) and Barthel Index scores. **Results:** Significant differences were observed between ALS and control groups (elevated muscle force and tension and interedge distance in non-ALS individuals). Between bulbar and spinal ALS subtypes, elevated values were observed in certain parameters in Bulbar ALS patients, indicating irregular vocal fold contact and weakened phonatory control, while spinal ALS exhibited increased values, suggesting higher phonatory muscle tension. Elevated biomechanical parameters were significantly correlated with low ALSFRS-R scores, suggesting a possible relationship between voice measures and functional decline. However, acoustic measurements showed no relationship with performance status. **Conclusions:** These results highlight the potential of voice analysis as a non-invasive, objective tool for monitoring ALS stage and differentiating between subtypes. Further research is needed to validate these findings and explore their clinical applications.

## 1. Introduction

Amyotrophic Lateral Sclerosis (ALS) is the most common form of motor neuron disease (MND), a group of neurodegenerative disorders affecting motor neurons. ALS specifically involves the degeneration of upper and lower motor neurons in the cortex, brainstem, and spinal cord. The point prevalence in Europe is estimated at 2.6–3 cases per 100,000 inhabitants, while in Spain, it reaches 5.4 cases per 100,000 inhabitants (twice the European average). The global incidence of ALS is estimated at 1.75 cases per 100,000 inhabitants per year, with an incidence of 1.59 cases per year in Europe and 1.4 cases per 100,000 inhabitants per year in Spain. Currently, ALS is the third most common neurodegenerative disease in terms of incidence, following dementia and Parkinson’s disease [1].

Studies analyzing voice characteristics in ALS have reported that, at the time of evaluation, approximately 70% of patients present a predominant spinal phenotype [1]. This progressive loss of motor function affecting bulbar control structures such as the mouth and larynx leads to speech disorders, including hypophonia, dysarthria, and anarthria [2]. In general, ALS manifests in two primary forms based on the site of symptom onset: spinal-onset ALS, where initial symptoms appear in the limbs, accounts for approximately 80% of cases, originates in the motor neurons of the spinal cord, initially affecting the limbs. In contrast, bulbar-onset ALS, primarily affecting speech and swallowing, represents around 20% of cases and arises from degeneration in the cranial nerve nuclei, leading to early impairment of bulbar musculature, which is essential for speech and swallowing. This distinction is relevant, as bulbar ALS is often associated with more rapid disease progression and greater impact on speech production, so patients with this variant generally have a poorer prognosis and reduced survival. Notably, dysarthria—characterized by impaired speech articulation—occurs in approximately 80% of ALS cases, irrespective of the site of disease onset [3]. It manifests as forced and slow speech production, short phrases, inappropriate pauses, articulatory imprecision, hypernasality, strained and tense voice, lower pitch, and reduced volume. Voice alterations may result from hyperabduction or abduction mechanisms, depending on whether the ALS subtype is predominantly bulbar or corticobulbar.

Several studies have demonstrated that dysphagia and voice disorders precede disease progression [3,4,5]. Acoustic and articulatory voice analysis have contributed to the early detection of degenerative changes, identifying voice alterations as potential biomarkers of disease progression. Thus, biomechanical voice analysis, unlike acoustic analysis and articulatory methods, provides an objective assessment of vocal fold dynamics and muscle coordination, offering additional insights into phonatory control [5]. This approach may facilitate early ALS diagnosis and assess disease progression, ultimately improving patient quality of life by delaying and/or reducing complications associated with disease progression (e.g., aspiration, sarcopenia, respiratory failure) while also complementing traditional analysis techniques for a more comprehensive voice assessment.

The objective of this study is to evaluate and correlate acoustic and biomechanical changes in the voices of ALS patients with their functional status and/or level of independence. This approach may facilitate early ALS diagnosis and assess disease progression, ultimately improving patient quality of life by delaying and/or reducing complications associated with disease progression, such as aspiration, sarcopenia, and respiratory failure. To explore this potential, we conducted an observational case series study focusing on ALS patients, incorporating a matched control group for descriptive comparison. Acoustic and biomechanical voice parameters were analyzed and correlated with functional status measures, aiming to identify meaningful associations that could support clinical decision-making and disease monitoring.

## 2. Materials and Methods

A descriptive observational case series study was conducted with an ALS cohort and a control group. The study was carried out at the Phoniatrics Department of Virgen Macarena Hospital in Seville, from June 2019 to January 2024. Patients were systematically selected through clinical records as they attended the consultation. The information was stored in a database using coded patient identifiers based on medical record numbers and NUHSA (Andalusian Unique Health Record Number). Only principal investigators had access to the database, which will be destroyed following data protection regulations (Organic Law 3/2018, December 5, on Personal Data Protection and Digital Rights Guarantee).

### 2.1. Participants

A total of 43 ALS patients (27 males and 16 females; age range 43–81 years) and 43 individuals with voice disorders, but without neurodegenerative diseases, were included in the initial phase of this study. The control group was matched for age and sex with the ALS cohort (27 males and 16 females; age range 43–80 years). The sample size was calculated using GRANMO software (version 8.0), with an α error set at 0.05 and a precision of ±0.2 in a two-tailed test, while accounting for a 10% follow-up loss rate. The calculation indicated that 5 subjects in the first group (ALS patients) and 15 in the second group (non-ALS control patients) would be needed to detect a statistically significant difference between two proportions: 0.8 (estimated prevalence of voice disorders in ALS patients) and 0.1 (estimated prevalence of voice disorders in the general population). These values were selected according to previously reported prevalence rates of voice alterations in ALS patients (~80%) and general populations (~1–17%) (Murtró, 2019; Lyberg-Ahlander, 2018) [6,7] with the Spanish Society of Otolaryngology (SEORL-CCC, 2017) [8] estimating a 7.7% prevalence in the general population. A 10% baseline prevalence was thus considered a reasonable approximation for the general population. However, it was considered more appropriate to increase the sample size. As a result, rather than requiring 3 controls per case, the controls were matched by age and sex.

All the process information was explained in the following schema (Figure 1).

### 2.2. Inclusion and Exclusion Criteria

Inclusion criteria for ALS patients required a definitive diagnosis according to the El Escorial Criteria [9], and being native Spanish speakers. Exclusion criteria included any other congenital or acquired neurological disorders besides ALS or pre-existing voice pathology.

For the control group, the inclusion criteria required native Spanish speakers with a non-neurological voice disorder. The exclusion criteria were the presence of known congenital or acquired neurological disorders.

### 2.3. Testing Procedures

#### 2.3.1. Data Acquisition

Voice samples were collected by recording a sustained vowel /a/ for a maximum of 4 s using a “Saramonic SR-XM1” (Saramonic International, Shenzhen, China) microphone connected to a mobile phone. The microphone was positioned 20–30 cm from the patient’s mouth; participants were instructed to maintain a constant distance from the microphone during phonation to ensure consistency in voice recordings. Recordings were conducted in a noise-free environment by an expert phoniatrician. The microphone was activated once phonation had begun to avoid contamination from silence or extraneous noise. Recording amplitude was maintained between 30% and 80%, monitored via a visual indicator.

#### 2.3.2. Biomechanical Voice Measures

Segments and specific points of the sustained vowel /a/ signal were analyzed to provide insights into vocal fold dynamics and detect pathology via radiated signals at the lips. The Voice Clinical Systems^®^ application (validated mobile software, version 1.4.0 for iOS, available exclusively for healthcare professionals) was used to generate an R3-type report (rapid alteration test). Table 1 summarizes the 22 biomechanical voice parameters assessed [10].

#### 2.3.3. Acoustic Voice Measures

The same sustained vowel /a/ recordings were analyzed using the Praat software (version 6.2.22) [11]. Three-second central segments were extracted and assessed via broadband spectrograms. Table 2 lists the acoustic parameters measured.

#### 2.3.4. Perceptual Voice Evaluation

The GRABS scale was used to assess perceptual voice parameters [12]. Table 3 provides details of the parameters evaluated and their scoring criteria.

#### 2.3.5. Functional and Dependency Measure

All ALS patients underwent Barthel Index and Amyotrophic Lateral Sclerosis Functional Rate Scale- Revised (ALSFRS-R) [13] assessments on the same day as voice sampling. The Barthel Index [14] evaluates dependence across 10 daily activities, with scores ranging from independent (100) to total dependence (0–20). The Barthel Index was used as a dependent variable to evaluate whether acoustic and biomechanical voice parameters could predict functional status in ALS patients. Control individuals, being voice patients without neurodegenerative disorders, were assumed to have full functional independence and thus assigned the maximum Barthel Index score. Table 4 summarizes the domains evaluated by ALSFRS-R.

### 2.4. Possible Confounding Variables in Voice Analysis

We investigated what voice parameters were influenced by sex and how voice parameters were modified within each sex. The results showed that sex was only associated with Fo and Pr1, which are voice parameters not relevant for ALS patients. This approach ensured that the observed differences in voice parameters were primarily attributable to ALS-related changes rather than inherent differences between male and female speakers.

Smoking habit was initially considered due to its effects on vocal function, such as inducing structural and functional alterations in the vocal folds, including edema and stiffness, which can impact voice parameters. However, after running a separate model to assess which parameters were influenced by smoking, we found no association between smoking and ALS-related voice features. Therefore, smoking was not included as a covariate in the final analysis.

### 2.5. Data Analysis

Descriptive analyses were performed using SPSS^®^ (version 29.0.2.0) [15] and R (version 3.3.0) [16]. Multivariate analyses were conducted using Korrigan Toolbox (version 0.1, Korrigan Sciences Ltd., Reading, UK) within MATLAB version 2015b (MathWorks, Natick, MA, USA) [17], a computational framework designed for advanced multivariate analysis. Matrix normalization was carried out using a median-based probabilistic quotient method (Dieterle et al., 2006) [18]. The analysis followed a two-step approach. Principal Component Analysis (PCA) was first applied for quality control and to explore the data structure, identifying four outliers and yielding a final sample of 39 ALS patients. This was followed by Orthogonal Projection to Latent Structures Discriminant Analysis (O-PLS-DA) (Bylesjö et al., 2007; Cloarec et al., 2005) [19,20], which enabled the identification of specific modulations driven by the factor of interest.

O-PLS-DA models were validated using seven-fold cross-validation, and model performance was evaluated based on the Q^2^Y value (goodness of prediction). Particular attention was given to inspecting the model score scatter plots (T) and cross-validated scores (Tcv) to detect and discard any signs of overfitting—models deviating from the diagonal were considered overfitted. Furthermore, model loadings plots were color-coded, where features most strongly associated with the discriminant component were shown in warm colors (closer to red), while those in cold colors (closer to blue) were not considered discriminant. Therefore, parameters in blue represent features with very low R2Y values, indicating minimal contribution to the model and a lack of statistical significance. As such, their directionality or potential biological relevance should not be interpreted, as they do not provide meaningful insights into group differences. Instead, the primary focus should be on parameters highlighted in warm colors, which exhibit stronger and more significant associations with the respective clinical index.

The O-PLS DA models were used to compare categorical patients’ groups as dependent variables (Y), including ALS vs. Control, Bulbar ALS vs. Control, Spinal ALS vs. Control, and finally Bulbar vs. Spinal ALS. Although O-PLS-DA is typically used for categorical outcomes, in this study, we extended the approach to continuous variables, specifically, the Barthel Index and ALSFRS-R, enabled by the flexibility of the Korrigan Toolbox. This approach preserved the full variability of functional scores and allowed for a more nuanced assessment of their relationship with voice parameters, avoiding arbitrary categorization. This methodology enables the capture of subtle associations between biomechanical and acoustic voice features and ALS functional status, ensuring a comprehensive and data-driven analysis.

The input variables (X) for all O-PLS-DA models were voice parameters extracted from the sample recordings.

To assess the sensitivity and specificity of individual voice parameters in differentiating ALS subtypes, we performed classical ROC curve analyses for individual biomarkers, as well as an overall O-PLS-DA classification analysis using features selected based on AUROC ranking. The model was trained using one latent variable for simplicity and generalizability.

## 3. Results

### 3.1. Study Population

A total of 39 ALS patients were analyzed after the exclusion of four outliers. Of these, 18 presented bulbar symptoms, primarily affecting speech and swallowing, while the remaining 21 exhibited mainly spinal symptoms (distal motor impairment) and reported some speech changes, according to the ALSFRS-R scale, which was administered to all participants. At diagnosis, 19 patients were smokers and 20 were non-smokers. Among the control group, five were smokers, while 38 were non-smokers.

Normal distribution of the sample was confirmed using the Shapiro-Wilk test for age (*p* = 0.42) and Chi-square test to assess whether the distribution of male and female participants deviates significantly from an expected 50% distribution sex (*p* = 0.17).

The sociodemographic and clinical characteristics of the participants are presented in Table 5.

### 3.2. Vocal Parameters Analysis

#### 3.2.1. Differences in Vocal Parameters Between ALS and Non-ALS Individuals

No significant differences were found between the total ALS group and the non-ALS group for either biomechanical or acoustic parameters.

In the OPLS-DA comparative model between patients with bulbar ALS and those without ALS, elevated values of Pr7 (percentage of the opening phase), Pr17 (associated with oedema observed during the closing phase), and Pr19 (correlate of the oedema observed during the closing phase) were observed for patients with bulbar ALS. In contrast, higher levels of Pr9 (effort required to reach and maintain maximal closure) and Pr12 (proportion of the glottal closure phase altered) were observed for non-ALS. The OPLS-DA models comparing bulbar ALS patients (*R^2^* = 0.16) and spinal ALS patients (*R^2^* = 0.07) to the control group were relatively weak, possibly due to the small sample size (Figure 2). Given the limited explanatory power of these models, additional statistical validation was performed using independent *t*-tests to assess the significance of individual parameters previously selected by OPLS-DA. While the analysis suggested changes in several Pr signals for bulbar ALS, only Pr12 showed a statistically significant difference between the groups (t (56.57) = 4.53, *p* = 3.08 × 10^−5^), with the application of the False Discovery Rate (FDR) correction to account for multiple comparisons. The 95% confidence interval ranged from 10.25 to 26.47. The mean Pr12 value was 21.27 for group 0 and 2.91 for group 1 (bulbar ALS). For spinal ALS, Pr21 did not reach statistical significance in the *t*-test (*p* > 0.05). Similarly, all other comparisons failed to achieve statistical significance, confirming that only Pr12 exhibited a robust difference in bulbar ALS patients.

On the other hand, no significant differences were found in GRABS scores between ALS and non-ALS groups, indicating that perceptual voice evaluation alone may not be sensitive enough to differentiate between these populations.

#### 3.2.2. Vocal Parameters and ALS Phenotypes (Bulbar vs. Spinal)

The O-PLS DA (*R*^2^ = 0.17) indicated significant differences between bulbar ALS and spinal ALS, with increased levels of Pr5 (percentage of time during which the vocal fold edges are approaching each other to achieve maximum glottic closure) and Pr18 (correlate of edema observed during the opening phase) in bulbar ALS, while spinal ALS was associated with higher levels of Pr9 (work required to achieve maximum closure and maintain it) (Figure 3). This finding was further confirmed by a *t*-test of the selected parameters, which showed significant differences for Pr5 (*p*-value = 0.047), Pr9 (*p*-value = 0.02), and Pr18 (*p*-value = 0.045). These differences remained significant after FDR correction. On the other hand, no significant differences were found in any of the assessed acoustic parameters.

#### 3.2.3. Evaluation of Voice Parameters as Biomarkers for ALS Subtype Differentiation

The ROC analysis demonstrated that individual voice parameters, such as Pr18 (AUC: 0.717), Pr9 (AUC: 0.688), and Pr5 (AUC: 0.682), showed moderate discriminative ability for ALS subtype differentiation (Figure 4). The overall O-PLS-DA model incorporating these features achieved an AUC of 0.71 (95% CI: 0.452–0.895), with sensitivity and specificity values varying across cross-validation iterations (panel m). The confusion matrix (panel n) and prediction accuracy distribution (panel o) further illustrate model performance.

### 3.3. Clinical Evaluation and Vocal Parameters

In the results of the O-PLS DA model using the Barthel Index, we observed that, regardless of sex or smoking status, ALS patients exhibited a significant relationship with Pr15 (degree of blockages in the total voice sample. A weaker association was also found with Pr14 (instability or inability to maintain amplitude across the entire voice sample) in individuals with ALS (Figure 5).

It is important to clarify that the Barthel Index is a numeric variable where 0 implies total dependence and 100 implies total independence, and the OPLS-DA model in this analysis was designed to capture the relationship of this numeric variable with vocal parameters. The classification between ALS and control individuals was not included as an outcome variable in this model. However, for ease of the figure interpretation, this includes labels indicating “ALS and control” based on known distributions of the Barthel Index (i.e., higher values associated with independence, and lower values associated with dependence).

These labels do not imply that a categorical classification was used in the model but rather help contextualize how vocal parameter variations align with different levels of functional impairment.

In addition, the O-PLS DA model, which compares the ALSFRS-R scores (R^2^ = 0.30) between controls and ALS patients, identified significant elevated levels of Pr14 (instability or inability to maintain signal amplitude considering the entire voice sample), Pr15 (degree of blockages in vibration considering the entire voice sample), and Pr16 (separation between the vocal cord edges) in the ALS group. The strongest correlation was observed for Pr15, with low ALSFRS-R scores. Conversely, the control group had significantly higher levels of Pr13 (instability or inability to withstand vibrational stress, considering a single vowel cycle) (Figure 6).

It is important to note that Figure 5 and Figure 6 are based on separate OPLS-DA models, with Figure 5 using the Barthel Index as the response variable and Figure 6 using ALSFRS-R. As a result, their Y-axis scales are not directly comparable. However, this does not affect the interpretation, as the key focus remains on the relative positioning of the parameters within each model.

As part of this study, we investigated the impact of smoking and sex on voice parameters exclusively in ALS patients to better understand their potential effects on the observed patterns. Acoustic vocal parameters (Jitter, Shimmer, and HNR) and biomechanical parameters (Pr2 to Pr22) did not show significant differences by sex among ALS patients, nor did they correlate with age or disease duration. However, Pr1 and F_0,_ which assess fundamental frequency, did exhibit significant differences. The comparative O-PLS DA model for the smoking factor was not particularly strong, but it suggested alterations in Pr20 levels (correlate of the edema observed during the opening phase) (Figure 7, Panel A). This potential influence was further examined using a *t*-test, which confirmed significant differences between the two groups (*p*-value = 0.04). As expected, non-smokers exhibited higher HNR levels, as an increase in HNR indicates that the non-smoking patient has a voice with greater clarity, smoothness, and less noise. On the other hand, sex-related differences were only observed in F_0_ and Pr1 (Figure 7B), which is consistent with the well-established distinctions between male and female voices under normal conditions.

Regarding the time since ALS diagnosis, weak but noteworthy variations were observed in Pr15 (degree of blockages in the vibration, considering the whole voice sample) and Pr16 (separation between the edges of the vocal folds). However, an expanded sample size would be necessary to confirm these findings conclusively.

## 4. Discussion

The deterioration of motor functions associated with ALS also encompasses speech alterations, with between 80% and 96% of individuals with this disease losing the ability to communicate verbally as it progresses [21]. Previous research has highlighted that voice analysis may serve as a key tool to identify bulbar dysfunction, as bulbar motor disorders (such as speech or swallowing difficulties) are often the first symptoms in 30% of ALS patients and manifest in all patients at more advanced stages of the disease [22].

The main objective of our study was to analyze the role of acoustic and biomechanical voice analysis to differentiate between the various types of ALS, with a particular focus on distinguishing bulbar from spinal forms and assessing the potential relationship between these parameters and the functional degree of the patient. The results revealed distinct biomechanical voice profiles between bulbar and spinal ALS, as well as a correlation between certain biomechanical parameters and lower scores on the ALSFRS and Barthel scales.

When comparing the group of patients with ALS to the group of patients without ALS, no clear correlation was observed in either acoustic or biomechanical voice parameters that would distinctly differentiate the ALS group. This contrasts with findings reported in other studies [22,23,24]. Such an absence of significant differences could be attributed to the heterogeneity of both the disease and the sample used, as patients without ALS had various voice pathologies. Additionally, the patient’s condition during the recordings may have influenced the results. It is worth noting that in previous studies, recordings were made in high, mid, and low pitches, with the high pitch being the most relevant for detecting alterations, whereas in our study, samples were exclusively taken in a mid or normal pitch. In this context, although comparing patients with ALS to those with other voice pathologies increases the variability of the sample, it also allows for more precise identification of voice features that are unique or particularly prominent in ALS patients. This approach highlights the specific differences of ALS, rather than merely observing a general pattern of vocal alteration that could be present in various pathologies. This result suggests that it may be necessary to conduct a more in-depth analysis by specific subgroups (bulbar and spinal ALS) to identify subtler alterations, although acoustic and biomechanical parameters may not be useful for detecting clear differences between individuals with ALS and those without ALS.

On the other hand, when comparing the bulbar ALS subgroup to non-ALS, notable differences were observed in Pr7, Pr17, and Pr19, parameters associated with glottal opening dynamics and an increase in the correlate of oedema in the glottic closure phase in patients with bulbar ALS. These differences could be related to the altered glottic mobility due to laryngeal muscle involvement that occurs in bulbar ALS. However, these differences were weak, and statistical significance was only found at the level of Pr12, which was found to be increased in patients without ALS, suggesting that vocal parameters may not be sensitive or specific enough to differentiate these groups clearly. The fact that patients with bulbar ALS present altered opening-related parameters and the correlate of glottal closure oedema reinforces the hypothesis that bulbar ALS primarily affects the opening phase and control of VC vibration. This difference could be useful for future diagnostic and follow-up studies, but further studies are needed to establish its clinical utility.

In relation to spinal ALS patients compared to those without ALS, significant differences were observed in Pr21, favoring the non-ALS group. This finding suggests a structural alteration, which is consistent with the fact that the non-ALS patients had voice pathologies. It also helps to evaluate how biomechanical analysis could differentiate between potential structural pathologies and neurodegenerative involvement.

In contrast, when comparing the bulbar and spinal ALS patient groups, an increase in the parameters P5, Pr9, and Pr18 was observed, suggesting that the two types of ALS affect vocal production differently. When comparing bulbar and spinal ALS patient groups, significant differences in vocal parameters were observed, suggesting distinct biomechanical adaptations in each phenotype. Specifically, in bulbar ALS, there is primarily an increase in parameters related to glottal opening and edema during the vocal fold opening phase (Pr5 and Pr18, respectively), which may be associated with slower articulation or more hypofunctional behavior (altered rhythm and prosody). On the other hand, spinal ALS patients exhibit greater compensatory behavior, characterized by increased effort in achieving and maintaining maximum glottal closure (Pr9), leading to a more hyperfunctional voice profile (affecting voice modulation). The significant differences in these parameters, supported by *t*-tests and FDR correction, strengthen the validity of these findings, suggesting they could be useful for distinguishing between ALS phenotypes in clinical practice. This compensatory mechanism may reflect adaptive strategies in response to motor neuron degeneration, which differ from the hypofunctional alterations typically observed in bulbar ALS. These distinct phonatory adaptations highlight the potential of biomechanical voice analysis in characterizing ALS subtypes, offering a complementary perspective to traditional clinical assessments. Additionally, this aligns with previous studies showing how clinical phenotypes significantly impact voice parameters, with articulatory alterations observed in bulbar ALS [22]. These findings reinforce the notion that the involvement of the upper motor neuron unit influences speech motor control [25].

Significant correlations were also identified between the biomechanical voice parameters and the total scores of the Barthel and ALSFRS-R functional scales. Regarding the Barthel Index, it was observed that lower scores were associated with higher values of Pr15 (degree of blockages) and a somewhat weaker association with Pr14 (instability or inability to maintain cycle amplitude). This suggests that patients with greater functional impairment also exhibit more pronounced alterations in vocal stability and the ability to maintain normal vibration. Likewise, patients with ALS and low ALSFRS-R scores showed elevated levels of Pr14, Pr15, and Pr16 compared to controls, particularly for Pr15. These findings, in conjunction with those observed regarding the Barthel Index, reinforce the idea that biomechanical voice parameters could serve as a useful prognostic tool for assessing functional disability and ALS progression. However, both the Barthel Index and the total ALSFRS-R score encompass a broader spectrum of functions, including limb weakness, and in the ALSFRS-R, respiratory capacity as well—factors that may not accurately reflect speech motor control in patients with bulbar-onset ALS. This multidimensionality, especially of the ALSFRS-R scale, implies that results should be interpreted with caution when used to predict survival in these patients [26]. Therefore, it might be of interest to separate the scores and include only the respiratory and/or oral domain scores (bulbar subscore) in the statistical analyses, as done by Milella et al. [23].

Regarding tobacco use, although the O-PLS DA model was not particularly strong, the differences observed in Pr20 and the higher HNR in non-smokers are significant for understanding how tobacco may impact vocal health and could complicate the interpretation of the results. Similarly, the differences in Pr1 and F_0_ between sexes, with values ranging from 105–139 Hz for males and 180–240 Hz for females, align with the well-known physiological differences [27].

Finally, in relation to the time elapsed since the diagnosis until the moment of the sample collection, a slight increase in the Pr15 (degree of blockages in the vibration considering the whole voice sample) and Pr16 (separation between the edges of the vocal folds) parameters was observed. However, it would be advisable to expand the sample size to confirm these findings. This phenomenon could indicate that the closing and opening phases of the vocal cords (VC) are altered or delayed, which may be associated with weakness or lack of motor control of the muscles responsible for regulating these movements. The elevation of these indices as the disease progresses could quantitatively reflect the progressive alteration of the biomechanical parameters of the voice, thus offering an objective tool to assess the severity of symptoms and the response to treatments or therapeutic interventions. In contrast, with respect to the acoustic parameters, no significant results were found, which differs from other studies that assessed patients at different stages of the disease, observing, for example, an increase in Jitter as the disease progresses. This finding suggests that it would be interesting to conduct evaluations at multiple time points throughout the course of the disease, which would allow for more data on its progression and contribute to a more detailed understanding of the associated acoustic changes.

Voice analysis is a promising, non-invasive tool for assessing speech motor deficits in ALS, with potential applications in early diagnosis and disease monitoring. Given that voice dysfunction often appears before other clinical symptoms, tracking biomechanical voice parameters may enable earlier intervention and better disease management. Additionally, integrating voice analysis into clinical practice could provide a quantitative measure of functional decline, supporting personalized therapeutic strategies and improving patient quality of life.

Despite the insights provided by this study, it is important to recognize some limitations. The small sample size may affect the generalizability of the findings, so a larger and more diverse sample would allow for more robust results and enable the analysis of subgroups according to disease progression. Additionally, as this is a cross-sectional study, it lacks longitudinal data to assess the evolution of vocal parameters and their relationship with clinical outcomes over time. Longitudinal studies would provide a more comprehensive understanding of these changes and their predictive value in ALS progression. Another important issue is that, being one of the first voice studies to use biomechanical analysis in ALS patients, we have not been able to compare our results with those of previous studies. Finally, cognitive decline was not assessed, which can significantly influence the production and modulation of the voice, affecting both motor aspects and those related to attention, memory, and prosody. While the motor component is the most recognized in ALS (affecting the control of muscles responsible for movement, including those involved in speech production), cognitive decline can also directly impact the production and perception of sounds [28]. This may result in difficulties in articulation, changes in rhythm and prosody, reduced attention and concentration capacity, impaired verbal memory, and alterations in emotional modulation. The interaction of these factors can complicate the precise diagnosis of vocal parameters in ALS patients, as well as make therapeutic interventions more complex [29].

## 5. Conclusions

In summary, voice analysis represents a promising, non-invasive, and objective tool for evaluating speech motor deficits in ALS. The acquisition of quantifiable measures of vocal characteristics, alongside traditional assessments, could contribute to a better understanding of disease progression, optimize clinical decision-making, and promote the development of specific interventions for patients. Ongoing research in this area can deepen our knowledge of ALS pathophysiology and contribute to improving its management, as well as the quality of life of those affected [29].

## Figures and Tables

**Figure 1 audiolres-15-00053-f001:**
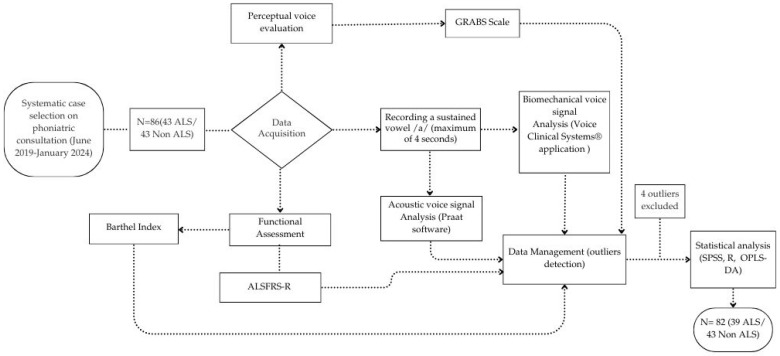
This schematic illustrates the flowchart employed in the study of acoustic and biomechanical changes in the voice of ALS patients, in relation to their functional status.

**Figure 2 audiolres-15-00053-f002:**
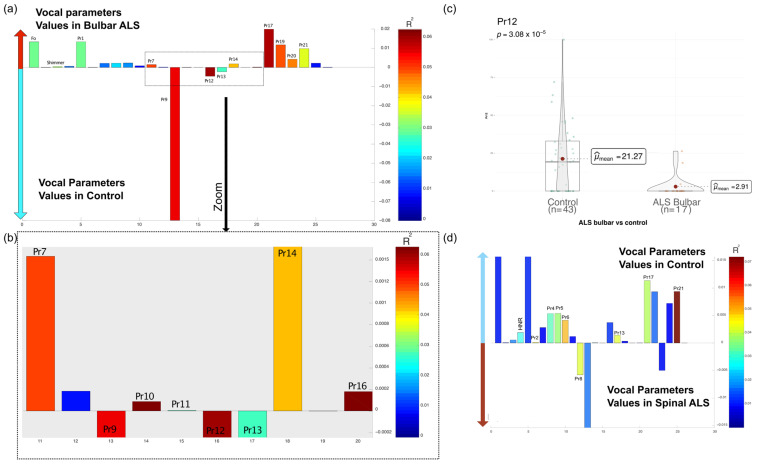
Differences in vocal parameters between ALS and non-ALS individuals. Loadings (bar plots) represent the contribution of each variable to the discriminant components and may vary across models depending on the data and components analyzed: (**a**) Bar plot showing differences in vocal parameters between bulbar ALS patients and control individuals. Warm colors indicate significant R^2^ values. Variables whose bars are higher in the positive Y-axis region are increased in the bulbar ALS group. Conversely, variables whose bars are higher in the negative Y-axis region are higher in the control. (**b**) Magnified view of the selected parameters from (**a**), emphasizing those with higher R^2^ values. In the OPLS-DA bar plots, the Y-axis represents the regression coefficients (weights) assigned to each vocal parameter by the model. These coefficients indicate the relative contribution of each parameter to the discrimination between groups (e.g., ALS subtypes vs. controls). Higher absolute values suggest a stronger influence of a given parameter on the separation, with positive and negative values reflecting differences in feature expression between the compared groups. Since OPLS-DA is a supervised multivariate method, the original vocal parameters are transformed into a latent space that maximizes inter-group variance while minimizing intra-group variance. The regression coefficients in the bar plots result from this transformation and provide insight into which features most effectively distinguish ALS subtypes from controls. Importantly, the scale of the Y-axis is model-dependent and varies according to the magnitude of the coefficients derived from each specific comparison. (**c**) Violin plot for Pr12, the only parameter showing a statistically significant difference in bulbar ALS patients compared to controls (*t*-test, *p*-value = 3.08 × 10^−5^). The boxplot within the violin plot represents the data distribution and central tendencies. (**d**) Bar plot comparing spinal ALS patients to control individuals, following the same color scheme as (**a**). Pr21 is higher in controls.

**Figure 3 audiolres-15-00053-f003:**
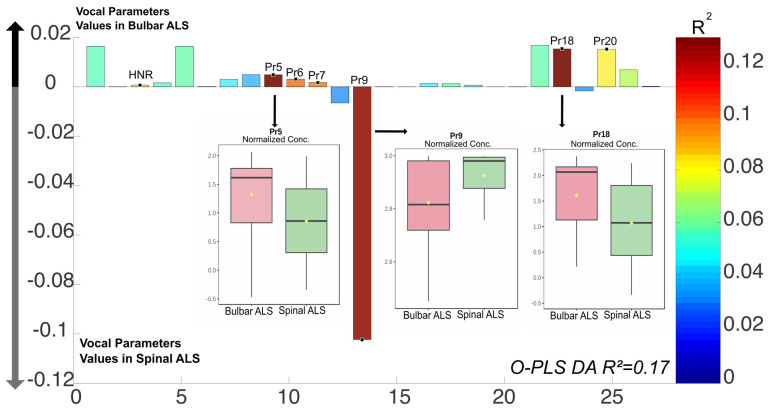
Differences in vocal parameters between bulbar ALS and spinal ALS patients: Bar plot showing the differences in vocal parameters between bulbar and spinal ALS patients based on an O-PLS DA model (R^2^ = 0.17). Warm colors indicate significant R^2^ values according to the O-PLS DA model. Values that are significantly higher in the bulbar ALS are plotted in the positive Y-axis region. Higher values in spinal ALS are plotted in the negative Y-axis region. Three parameters (Pr5, Pr9, and Pr18) demonstrated significant differences between the two groups and are highlighted with boxplots representing their distributions. Boxplots show the normalized value of each parameter in bulbar ALS (pink) and spinal ALS (green), with yellow dots indicating mean values. Pr5 and Pr18 were significantly higher in bulbar ALS, whereas Pr9 was significantly higher in spinal ALS (*p*-values = 0.047, 0.02, and 0.045, respectively, after FDR correction).

**Figure 4 audiolres-15-00053-f004:**
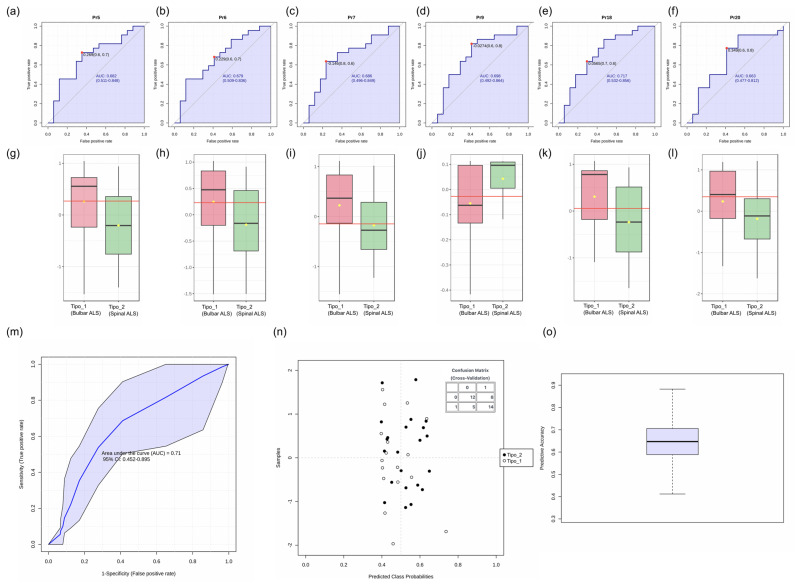
ROC curve analysis and classification performance for ALS subtype differentiation based on voice parameters: (**a**–**f**) Classical ROC curve analysis for individual voice biomarkers (Pr5, Pr6, Pr7, Pr9, Pr18, Pr20). The ROC curves illustrate the true positive rate (sensitivity) versus the false positive rate (1-specificity) for each parameter. The AUC (area under the curve) values indicate the discriminative ability of each biomarker, with confidence intervals shown in parentheses. The red dot marks the optimal cutoff point based on the Youden Index. (**g**–**l**) Boxplots comparing selected voice biomarkers between bulbar ALS (pink) and spinal ALS (green) subtypes. The horizontal red line represents the median, while the yellow dot indicates the mean value for each group. (**m**) ROC curve of the O-PLS-DA classification model using the most discriminative features (Pr18, Pr9, Pr5). To generate a smooth curve, 100 cross-validations were performed, and the results were averaged. The shaded area represents the 95% confidence interval for the ROC curve, with an overall AUC of 0.71 (95% CI: 0.452–0.895). (**n**) Scatter plot of predicted class probabilities for bulbar and spinal ALS samples. The confusion matrix (inset) shows the classification results from cross-validation, indicating the number of correctly and incorrectly classified samples for each ALS subtype. (**o**) Boxplot of the prediction accuracy across cross-validation iterations. The median and interquartile range of model performance are displayed, highlighting variability in classification accuracy.

**Figure 5 audiolres-15-00053-f005:**
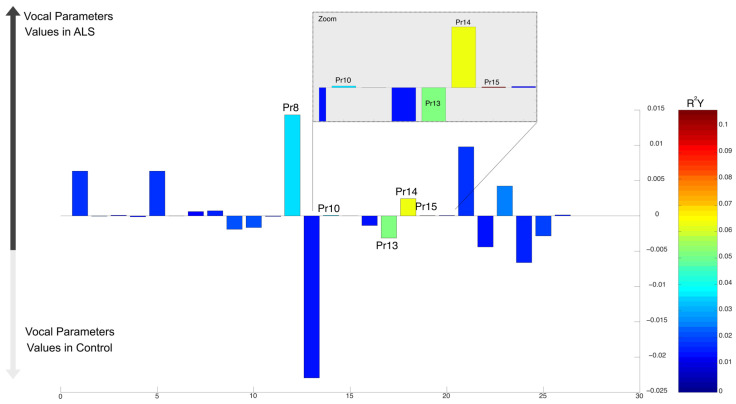
Differences in vocal parameters associated with the Barthel Index in ALS patients and control individuals: Bar plot illustrating the differences in vocal parameters between ALS patients and control individuals based on an O-PLS DA model (R^2^ = 0.30). Warm colors indicate significant differences in the parameter of interest. When the bar is increased in the positive region of the Y axis, it means that the value is higher in the ALS group. Conversely, when the bar is increased in the negative region of the Y axis, the parameter is increased in the control group. A zoomed-in section highlights specific parameters with notable differences. Pr15 was significantly higher in ALS patients (in warm, intense color).

**Figure 6 audiolres-15-00053-f006:**
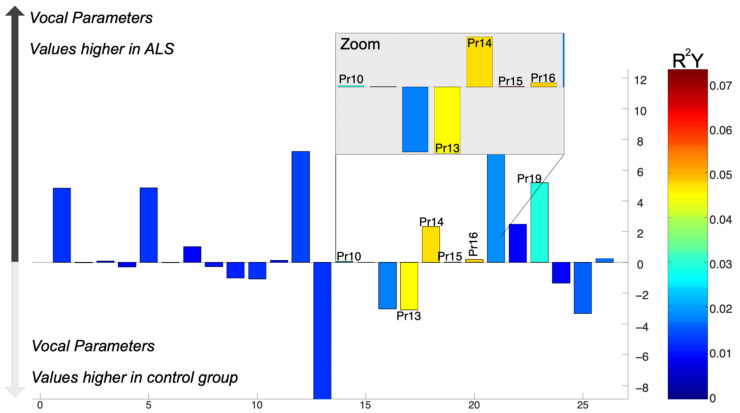
Differences in vocal parameters associated with ALSFRS-R scores in ALS patients and control individuals: Bar plot illustrating the differences in vocal parameters between ALS patients and control individuals based on an O-PLS DA model (R^2^ = 0.30). As with the analysis presented in Figure 5, the OPLS-DA model used for Figure 6 was designed to assess the relationship between vocal parameters and ALSFRS-R scores within the ALS group, rather than to classify ALS versus control individuals. Significant R^2^ values are colored in warm colors (i.e., red to yellow). The direction of the bars in Figure 6 does not indicate a positive or negative correlation but rather represents the measured vocal parameter values. Parameters plotted in the positive Y-axis region correspond to ALS patients, while those plotted in the negative Y-axis region correspond to controls, as indicated.

**Figure 7 audiolres-15-00053-f007:**
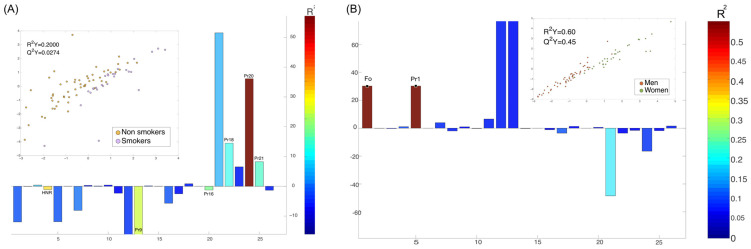
Influence of smoking and sex on vocal parameters in ALS patients: (**A**) O-PLS DA model assessing the effect of smoking on vocal parameters (R^2^Y = 0.20). The model suggests alterations in Pr20 levels, with smokers exhibiting significantly higher levels. (**B**) O-PLS DA model evaluating sex-related differences in vocal parameters (R^2^Y = 0.60). Significant differences were observed only for F_0_ and Pr1. The warm color bars indicate the R^2^Y values are significant, with positive values representing higher levels in the women’s group.

**Table 1 audiolres-15-00053-t001:** Biomechanical Voice Parameters Assessed.

Parameter	Description	Clinical Relevance
Pr1	Fundamental frequency. Normal range: 180–240 Hz (females)/105–139 Hz (males)	Indicator of pitch control and vocal fold vibration stability
Pr2	Ratio of cycles in the closing phase between free edges	Indicates regularity in vocal fold closure
Pr3	Increased open phase and asymmetry in free edge vibration	May be linked to vocal fold asymmetry and phonatory instability
Pr4	Percentage of time spent in the approximation phase for glottal closure	Affects vocal efficiency and closure coordination
Pr5	Percentage of time vocal folds remain separated	May indicate hypofunctional phonation
Pr6	Duration of the opening phase (vocal fold separation)	Associated with airflow dynamics during phonation
Pr7	Duration of the opening phase (vocal fold approximation)	Reflects closure speed and coordination
Pr8	Tension associated with glottal closure (hyperfunctional glottis)	Suggests excessive muscular tension affecting phonation
Pr9	Glottal closure force	May indicate compensatory mechanisms or phonatory inefficiency
Pr10	Index of optimal energy use during voice production	Evaluates phonatory efficiency and voice sustainability
Pr11	Incomplete glottal closure (gap between edges)	May contribute to breathy voice quality or phonatory inefficiency
Pr12	Alteration in the proportion of the closing phase (GAP size)	Can reflect structural or functional glottal insufficiency
Pr13	Instability in maintaining vibratory tension within a single vocal cycle	Linked to phonatory instability and tremor
Pr14	Instability in maintaining vibratory tension within a single vocal cycle	Similar to Pr13, related to vocal fatigue and tremor
Pr15	Degree of vibratory blockages throughout the voice sample	May indicate phonatory spasms or irregular vocal fold movement
Pr16	Separation between the edges of the vocal folds	Reflects glottal configuration and closure patterns
Pr17	Mucosal wave correlate observed during the closing phase	Related to vocal fold pliability and phonatory function
Pr18	Mucosal wave correlate observed during the opening phase	Provides insight into vibratory mechanics
Pr19	Edema correlate observed during the closing phase	Suggests possible inflammation or structural changes
Pr20	Edema correlate observed during the opening phase	Related to phonatory effort and tissue properties
Pr21	Structural imbalance index	Indicates potential vocal fold asymmetry or stiffness
Pr22	Correlates with potential structural alterations in the free edge mass or supraglottic structures	May indicate vocal fold lesions or supraglottic hyperfunction

**Table 2 audiolres-15-00053-t002:** Acoustic Voice Parameters Assessed.

Parameter	Description	Normal Range
Fundamental frequency	Fundamental frequency (acoustic analysis, Praat). Same normative values as Pr1 but may slightly differ due to measurement method	180–240 Hz (females)/105–139 Hz (males)
Shimmer	Amplitude perturbation	0–3%
Jitter	Fundamental frequency variation between cycles	~1%
Harmonics to Noise Ratio (HNR)	Signal to noise measure	Values < 20 indicate dysphonia

**Table 3 audiolres-15-00053-t003:** GRABS Scale Parameters.

Parameter	Description	Scoring (0 = Normal, 3 = Severe)
Global	Global voice impairment	0 to 3
Roughness	Irregular glottic mobility	0 to 3
Asthenia	Phonatory weakness	0 to 3
Breathiness	Turbulence due to glottic insufficiency	0 to 3
Strain	Excessive effort with hyperkinesia	0 to 3

Note: Dysphonia severity was categorized as mild (0–3), moderate (4–6), severe (7–9), and very severe (10–15).

**Table 4 audiolres-15-00053-t004:** ALSFRS- R Functional Domains.

Domain	Description	Scoring (0 = Complete Impairment, 4 = Normal Function)
Bulbar	Speech, salivation and swallowing	0 to 4
Fine motor skills	Writing, utensil use and hygiene/dress	0 to 4
Gross motor skills	Turning in bed, walking and stair climbing	0 to 4
Respiratory functions	Dyspnea, orthopnea and ventilatory support	0 to 4

Note: This comprehensive evaluation provided key insights into the functional and dependency status of ALS patients in the study.

**Table 5 audiolres-15-00053-t005:** Sociodemographic and clinical characteristics.

Sociodemographic	Clinical Score ALS	Clinical Score Non-ALS
Age	63.31 ± 9.45	64.02 ± 7.47
N (men/women)	39 (24/15)	43 (27/16)
Smoking (No/Yes)	20 (24.4%)/19 (23.2%)	35 (81.4%)/8 (18.6%)
Time to diagnosis ALS (months)	20.56 ± 24.15	Not applicable
Type of ALS: Bulbar/Spinal	17 (20.7%)/22 (26.8%)	Not applicable
ALSFRS-R	35.38 ± 6.98	48 ± 0
Barthel	67.85 ± 20.95	100 ± 0
GRABS	3.92 ± 4.15	5.7 ± 2.25

Note: Mean ALSFRS-R score: Corresponding to advanced intermediate stage; Mean Barthel score: Corresponding to moderate dependency; Mean GRABS score: Corresponding to moderate levels of dysphonia.

## Data Availability

The data that support the findings of the study are available from the corresponding author upon reasonable request.

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
