# Peer review of "Relationship Between Voice Analysis and Functional Status in Patients with Amyotrophic Lateral Sclerosis"

_audiolres, 2025, doi:10.3390/audiolres15030053_

Round 1

Reviewer 1 Report

Comments and Suggestions for Authors

Summary: In the manuscript, Pérez-Bonilla et al. compared biomechanical and acoustic voice analysis results between patients with ALS and non-neurodegenerative voice disorders, as well as between bulbar and spinal ALS. Functional status was evaluated using the Barthel Index and ALSFRS-R scores, and their correlations with the voice analysis results were evaluated using OPLS-DA. The results highlight the possibility of using voice analysis to differentiate different ALS subtypes and to monitor the functional status in ALS patients.

Overall assessment: The manuscript is overall nicely written, and the research topic fits well with the scope of the journal. My comments and suggestions are primarily on the clarity of the statistical results, including the justification of methods, details of the stats values, and data visualization.

Page 1, Abstract: The names of the biomechanical parameters “Pr#” appeared multiple times in the abstract, but it does not give useful information without referencing Table 1. I would recommend either remove these parameter names from the abstract or replace them with specific descriptions (i.e., the descriptions in the second and third columns of Table 1).  

Page 1, line 40: “ALS progression” suggests a longitudinal change across time, while the present study is a cross-sectional study. I would recommend replacing “progression” with more static descriptions, such as severity or stage.

Page 1, line 40: “differentiating between subtypes” – while the analysis in the manuscript identified voice parameters that differ between bulbar and spinal subtypes, further analysis is needed to justify the potential of using these parameters to differentiate between subtypes. Specifically, the sensitivity and specificity of predicting the subtypes from the voice parameters in the test set of the seven-fold cross validation (line 166) could be reported, especially for the bulbar vs. spinal ALS categorization.

Page 2, lines 49-50: What’s the time window for these incidence data? Per year?

Page 2, lines 54-60: The definition and proportion of the spinal and bulbar subtypes are not very clear. Specifically, is the “spinal-onset ALS” in line 58 the same as “spinal ALS” in line 54? If so, the percentages (70% vs. 80%) are not consistent in these two descriptions. If they are different, a couple sentences could be added to explain the difference and to clarify which definition is used to categorize the patients in the present study.

Page 2, line 73: The introduction of biomechanical voice analysis could be better motivated. Specifically, how does it complement the acoustic/articulary voice analysis methods, and what additional benefit it might offer for ALS monitoring compared to the acoustic/articulary analysis alone?

Page 2, line 84: Since there is a control group, I was wondering whether “case-control study” might be a better description of the study design than “case series study”.

Page 3, line 102: does the “preliminary study” refer to the study reported in the main body of the manuscript? If so please consider replacing with “the current study” or “the present study”. Otherwise, it might lead the readers to think that there is another preliminary study conducted prior to the study reported in the manuscript.

Page 3, lines 107-108: What variables do the “two proportions” correspond to? The rest of analysis doesn’t seem to involve comparing proportions across groups, so the method for power analysis might need further explanation and justification.

Page 3, Figure 1: The placement of “4 outliers excluded” might need adjusting. From the text, it seems that the step of “4 outliers excluded” was performed before the multivariate analysis.

Page 4, line 123: Since the present study is observational and not experimental, the heading of this section could be changed to “Procedures” or “Testing Procedures”.

Page 4, line 127: It’s not very intuitive what “with phonation adapted to maintain this distance” mean – did the distance automatically change based on the speech response?

Page 4, Table 1: If possible, it could be helpful to describe each parameter separately (e.g., what’s the difference between Pr2 and Pr3?)

Page 5, Table 2: Since F0 appeared both in Table 1 (biomechanical parameters) and Table 2 (acoustic parameters), I was wondering whether F0 was included once or twice in the subsequent statistical models? From the figures it seems that both F0 and Pr1 were included in the models, which would be a duplicate of the same parameter.

Page 5, Table 3: Were the GRABS scale parameters used for the subsequent analysis? It doesn’t seem to be included in the figure or the descriptions. If GRABS data were collected but not used in the analysis, please include a justification for this choice.

Page 6, line 160: Korrigan Toolbox is a less known stats tool compared to SPSS and R. It would be great if a citation (of a peer-reviewed paper) and/or a hyperlink to the website could be added to show the credibility of the toolbox.

Page 6, line 163: Please spell out the full name of OPLS here since it’s the first appearance and it’s not spelled out anywhere else in the main body of the paper.

Page 6, line 164: OPLS is usually used for categorical outcomes, but Barthel Index and ALSFRS-R are continuous outcome variables. It would be very helpful if a few sentences could be added here to explain how OPLS was performed on the continuous variables. For example, did the authors transform the Barthel Index and ALLSFRS-R into categorical variables by dividing the patients into a high functionality group and a low functionality group?

Also, from the results section, it seems that the patient groups (ALS vs. Control, Bulbar ALS vs. Control, Spinal ALS vs. Control, and Bulbar vs. Spinal) also serve as the outcome Y variable in OPLS.

Page 6, line 174: It could be helpful to add a sentence to justify why including smoking habit in the analysis. Specifically, to what extent is smoking relevant to voice features, voice disorders, and ALS?

Page 6, line 177: Sex is a categorical data and therefore cannot be described as “normally distributed” or tested for normal distribution. Might worth considering replacing it with a chi-square test if the authors would like to check whether the proportions of female and male patients are significantly deviant from 50%.

Page 7, lines 189-191: How was the lack of significant differences determined? Specifically, was it based on the OPLS-DA analysis, or t-tests? It could be helpful to include the stats values here to support the lack of significant differences.

Page 7, lines 193-195: What’s the criteria of parameter selection – What’s the cut-off R^2 value? It’s described as “Warm colors” in the caption of Figure 2 but that’s vague. Since there are R2Y and R2X values in OPLS, it could also be helpful to add a sentence to describe what R2 values here means specifically.

Page 7, line 198: Were the results of t-tests corrected for multiple comparisons (by using FDR or Bonferroni correction etc)? Also please include t value and df when reporting the result of t-tests.

Page 7, Figure 2: What is the unit of the vertical axis? The scale of the vertical axis is different between panels a) and b), making it confusing. This applies to the other figures as well, especially figures 5 & 6 where the range of the vertical axis differs drastically from the other figures.

The vertical directions of panels a) and d) are swapped – it might be more straightforward if the downward arrow corresponds to the control group in both panels.

The font size in panel c) needs to be increased.

Page 8, lines 238-239: Based on this paragraph, it seems that the relationship between Barthel Index and voice analysis was only assessed in ALS patients. However, Figure 4 suggests both ALS and the control group were included in the analysis. I was wondering which one is the accurate description of the analysis. If both groups were included, was the Barthel Index treated as the only outcome variable (regardless of whether the patients are from the ALS or control group), or was the “ALS vs. control” included somewhere in the analysis as a predictor/covariate?

Page 8, line 239: “the elevated value of Pr15” – this doesn’t seem to be consistent with Figure 4. The bar for Pr15 is barely visible even after zooming in. (To clarify, the current figure is not necessarily wrong, since it could be that Pr15 is comparable between ALS and control group but showed a significant correlation with Barthel Index only in the ALS group. It’s just not a very effective way to illustrate the results describe in the text.)

Page 9, Figure 4: It might be helpful to modify the figure or add a panel to show which parameters are positively and negatively corelated with Barthel Index, in the same format as the current panel. This also applies to ALSFERS-R scores in Figure 5.

I was also confused by the difference in the height and directions of the bars in Figures 4 and 5. If both figures are showing the differences in vocal parameters between ALS and control groups, as described in the figure captions, they should have the same height and direction in Figures 4 and 5, only different colors representing different R2 values. However, currently they are very different especially for Pr12-21.

Page 10, Figure 6: Were the sex and smoking analyses performed on all patients, or ALS patients only?

Also, it is unclear which smoking habit “F0/F1” refers to, and which sex “Sexo1/Sexo2” refers to. Please replace the labels in the legend with the actual description of the group e.g., “Smoker/Non-smoker”.

Page 11, line 327: Pr9 and Pr12 also showed differences, just in the opposite direction.

Page 12, line 344: Pr5 and Pr18 showed increase in the bulbar group, while Pr9 showed decrease in the bulbar group, compared to the spinal group.

Author Response

Response to Reviewer 1 Comments

Dear reviewer,

We appreciate the time and dedication you have put into reviewing our manuscript and we will now respond to your comments point by point, which will undoubtedly improve its quality.

Comment 1: Page 1, Abstract: The names of the biomechanical parameters “Pr#” appeared multiple times in the abstract, but it does not give useful information without referencing Table 1. I would recommend either remove these parameter names from the abstract or replace them with specific descriptions (i.e., the descriptions in the second and third columns of Table 1). 

Response 1: Thank you for your valuable suggestion. We acknowledge that the repeated use of biomechanical parameter labels (Pr#) in the abstract may not provide sufficient clarity without direct reference to Table 1.

To enhance readability while maintaining the integrity of our findings, we have removed the parameter names that do not alter the meaning of the text and have only replaced Pr8 and Pr16 with their corresponding descriptive terms to ensure the text remains coherent.

We appreciate your feedback, which has helped us improve the clarity and accessibility of our abstract. This revision can be found on Page 1, Line 32 “[…] (elevated muscle force and tension and inter- edge distance in non- ALS individuals)” replacing “[..] Pr8 and Pr16”.

Comment 2: Page 1, line 40: “ALS progression” suggests a longitudinal change across time, while the present study is a cross-sectional study. I would recommend replacing “progression” with more static descriptions, such as severity or stage.

Response 2: Thank you very much for your comment. We acknowledge that the term “ALS progression” may imply a longitudinal change over time, which is not appropriate for our cross-sectional study. To ensure accuracy, we have replaced “progression” with “stage” throughout the manuscript.

The modified text on Page 1, line 41 now reads: "[...] to better understand ALS stage and its association with biomechanical voice parameters."

This modification aligns the terminology with the cross-sectional nature of the study and avoids any unintended implications of longitudinal analysis.

Comment 3: Page 1, line 40: “differentiating between subtypes” – while the analysis in the manuscript identified voice parameters that differ between bulbar and spinal subtypes, further analysis is needed to justify the potential of using these parameters to differentiate between subtypes. Specifically, the sensitivity and specificity of predicting the subtypes from the voice parameters in the test set of the seven-fold cross validation (line 166) could be reported, especially for the bulbar vs. spinal ALS categorization.

Response 3: Thank you for your insightful comment regarding the sensitivity and specificity of voice parameters for ALS subtype differentiation. We appreciate the opportunity to clarify this aspect of our study. Our study was initially designed to identify statistically significant differences in voice parameters between ALS subtypes (bulbar vs. spinal) rather than to develop a classification model. Therefore, we did not originally include sensitivity and specificity analysis, as our primary focus was on group comparisons rather than predictive classification.

However, based on your feedback, we have now conducted a sensitivity and specificity analysis using Receiver Operating Characteristic (ROC) curve methodology. The results of individual biomarker analyses are presented in panels (a)–(f), while panel (m) shows the aggregated ROC analysis using the O-PLS-DA algorithm with one latent variable, selecting Pr18, Pr9, and Pr5 as the most discriminative features. The area under the curve (AUC) was 0.71 (95% CI: 0.452–0.895), indicating a reasonable discriminative ability. The confusion matrix from cross-validation is presented in panel (n), while panel (o) provides the distribution of prediction accuracy across cross-validation iterations.

We have added the following details to the revised manuscript:

Methods Section Page 8 Lines 232- 236: "To assess the sensitivity and specificity of voice parameters in differentiating ALS subtypes, we performed a classical ROC curve analysis for individual biomarkers and an overall classification analysis using an orthogonal partial least squares discriminant analysis (O-PLS-DA) model. The model was trained with one latent variable and features selected based on AUROC ranking.”

Results Section Page 11 Lines 323-329: "The ROC analysis demonstrated that individual voice parameters, such as Pr18 (AUC: 0.717), Pr9 (AUC: 0.688), and Pr5 (AUC: 0.682), showed moderate discriminative ability for ALS subtype differentiation (panels a–f). The overall O-PLS-DA model incorporating these features achieved an AUC of 0.71 (95% CI: 0.452–0.895), with sensitivity and specificity values varying across cross-validation iterations (panel m). The confusion matrix (panel n) and prediction accuracy distribution (panel o) further illustrate model performance."

Figure 4 in Page 12. Legend figure 4 Page 12, Lines 331- 346: “ROC curve analysis and classification performance for ALS subtype differentiation based on voice parameters. (a–f) Classical ROC curve analysis for individual voice biomarkers (Pr5, Pr6, Pr7, Pr9, Pr18, Pr20). The ROC curves illustrate the true positive rate (sensitivity) versus the false positive rate (1-specificity) for each parameter. The AUC (area under the curve) values indicate the discriminative ability of each biomarker, with confidence intervals shown in parentheses. The red dot marks the optimal cutoff point based on the Youden Index. (g–l) Boxplots comparing selected voice biomarkers between bulbar ALS (pink) and spinal ALS (green) subtypes. The horizontal red line represents the median, while the yellow dot indicates the mean value for each group. (m) ROC curve of the O-PLS-DA classification model using the most discriminative features (Pr18, Pr9, Pr5). To generate a smooth curve, 100 cross-validations were performed, and results were averaged. The shaded area represents the 95% confidence interval for the ROC curve, with an overall AUC of 0.71 (95% CI: 0.452–0.895). (n) Scatter plot of predicted class probabilities for bulbar and spinal ALS samples. The confusion matrix (inset) shows the classification results from cross-validation, indicating the number of correctly and incorrectly classified samples for each ALS subtype. (o) Boxplot of the prediction accuracy across cross-validation iterations. The median and interquartile range of model performance are displayed, highlighting variability in classification accuracy.”

Comment 4: Page 2, lines 49-50: What’s the time window for these incidence data? Per year?

Response 4: Thank you for your valuable comment. The prevalence values reported (2.6–3 cases per 100,000 inhabitants in Europe and 5.4 cases per 100,000 inhabitants in Spain) represent point prevalence, which refers to the number of existing cases at a given time, rather than an annual rate. To clarify this, we have revised the text as follows: "The point prevalence in Europe is estimated at 2.6–3 cases per 100,000 inhabitants, while in Spain, it reaches 5.4 cases per 100,000 inhabitants (twice the European average)."

This revision, found on Page 2, lines 50- 52, explicitly specifies that the data refer to point prevalence.

Comment 5: Page 2, lines 54-60: The definition and proportion of the spinal and bulbar subtypes are not very clear. Specifically, is the “spinal-onset ALS” in line 58 the same as “spinal ALS” in line 54? If so, the percentages (70% vs. 80%) are not consistent in these two descriptions. If they are different, a couple sentences could be added to explain the difference and to clarify which definition is used to categorize the patients in the present study.

Response 5: Thank you very much for your comment. We acknowledge the need for clarification regarding the terminology and reported percentages for spinal and bulbar ALS subtypes. The distinction between spinal-onset ALS and spinal ALS has now been explicitly clarified in the text.

Additionally, we have modified Page 2, lines 57- 59 to ensure consistency in the reported percentages. The text now reflects that while approximately 80% of ALS cases are classified as spinal-onset ALS at disease onset, studies on voice characteristics in ALS have reported that, at the time of evaluation, around 70% of patients exhibit a predominant spinal phenotype, while 30% present a bulbar phenotype.

We sincerely appreciate your insightful suggestion, which has allowed us to refine the introduction, ensuring alignment with the existing literature and enhancing clarity in the classification criteria used in voice studies.

Comment 6: Page 2, line 73: The introduction of biomechanical voice analysis could be better motivated. Specifically, how does it complement the acoustic/articulary voice analysis methods, and what additional benefit it might offer for ALS monitoring compared to the acoustic/articulary analysis alone?

Response 6: Thank you for your constructive comment. We acknowledge the need to better justify the role of biomechanical voice analysis in the introduction. To address this, we have modified lines 81-85 on page 2 to explicitly describe how biomechanical voice analysis complements acoustic and articulatory methods: “Thus, biomechanical voice analysis, unlike acoustic analysis and articulatory methods, provides an objective assessment of vocal fold dynamics and muscle coordination, offering additional insights into phonatory control.” And completing with the following explanatory sentence at the end of the existing text on Page 2 line 88: “[…] while also complementing traditional analysis techniques for a more comprehensive voice assessment.”

Specifically, we have highlighted that acoustic and articulatory analysis focus on the perceptual and spectral characteristics of speech, whereas biomechanical analysis provides an objective assessment of vocal fold dynamics, muscle coordination, and phonatory control. This allows for a more comprehensive evaluation of voice function in ALS, offering potential advantages for early detection and disease monitoring.

This revision clarifies the unique contribution of biomechanical voice analysis and its relevance to ALS assessment.

Comment 7: Page 2, line 84: Since there is a control group, I was wondering whether “case-control study” might be a better description of the study design than “case series study”.

Response 7: Thank you for your valuable comment. While our study includes a control group for comparison, it is not a case-control study in the traditional epidemiological sense, as no exposure-outcome relationship is being assessed, nor are cases and controls selected based on prior exposure to a risk factor. Instead, this study is descriptive and observational, aiming to characterize voice parameters in ALS patients compared to individuals with non-neurological voice disorders.

To ensure clarity, we have revised Page 3, line 95- 97, explicitly stating that this is a descriptive case series study with a matched control group rather than a case-control study:” To explore this potential, we conducted an observational case series study focused on ALS patients, incorporating a matched control group for descriptive comparison.”

Comment 8: Page 3, line 102: does the “preliminary study” refer to the study reported in the main body of the manuscript? If so, please consider replacing with “the current study” or “the present study”. Otherwise, it might lead readers to think that there is another preliminary study conducted prior to the study reported in the manuscript.

Response 8: Thank you for your insightful comment. The term “preliminary study” was used to indicate that data collection was still ongoing and that additional patients were being recruited for further analysis. However, we acknowledge that this wording might suggest a separate prior study rather than an ongoing process.

To clarify this, we have revised Page 3, line 113, replacing “preliminary study” with “the initial phase of this study”. This ensures that readers understand that the results presented correspond to the current dataset, while future analyses may expand upon these findings as new data become available.

Comment 9: Page 3, lines 107-108: What variables do the “two proportions” correspond to? The rest of analysis doesn’t seem to involve comparing proportions across groups, so the method for power analysis might need further explanation and justification. 

Response 9: We appreciate your comment regarding the justification of the two proportions used in the power analysis. The proportions refer to the estimated prevalence of voice disorders in ALS patients (80%) and in the general population without neurological disorders (10%).

The 80% prevalence in ALS patients is supported by evidence indicating that dysphonia is a frequent symptom in ALS, especially in bulbar-onset cases where voice impairments are among the earliest manifestations. According to the Regional Association of People Affected by Amyotrophic Lateral Sclerosis (Asociación Regional de Afectados de Esclerosis Lateral Amiotrófica, ARAELA), voice alterations are a common feature in ALS progression (ARAELA).

The 10% prevalence of voice disorders in the general population is within the range reported in previous epidemiological studies. Some studies estimate that 1%–17% of the general population has experienced voice disorders (Murtró, 2019; Lyberg-Ahlander, 2018), while others indicate that the prevalence can be higher in specific subpopulations, such as teachers, where up to 20% present voice disorders (Hernández, 2020).

Additionally, according to the Spanish Society of Otolaryngology and Head and Neck Surgery (SEORL-CCC, 2017), approximately 7.7% of the general population (equivalent to 1 in 13 people) experience voice disorders, yet most cases go untreated.

Based on this evidence, a 10% baseline prevalence was considered a reasonable approximation for the general non-neurological population.

To clarify this point, we have revised the Methods section (Page 3, lines 119-126) to explicitly define that the two proportions correspond to the prevalence of voice disorders in ALS patients and in the general population, along with references supporting these estimations.

“[…] 0.8 (estimated prevalence of voice disorders in ALS patients) and 0.1 (estimated prevalence of voice disorders in the general population). These values were selected according to previously reported prevalence rates of voice alterations in ALS patients (~80%) and general populations (~1–17%) (Murtró, 2019; Lyberg-Ahlander, 2018) with the Spanish Society of Otolaryngology (SEORL-CCC, 2017) estimating a 7.7% prevalence in the general population. A 10% baseline prevalence was thus considered a reasonable approximation for the general population”

Comment 10: Page 3, Figure 1: The placement of “4 outliers excluded” might need adjusting. From the text, it seems that the step of “4 outliers excluded” was performed before the multivariate analysis.

Response 10: We appreciate your observation regarding the placement of “4 outliers excluded” in Figure 1. Indeed, the exclusion of outliers was conducted before the multivariate analysis to ensure data integrity. To accurately reflect this methodological step, we have adjusted the placement of “4 outliers excluded” in Page 4, Figure 1 to correctly represent its occurrence before statistical modeling. Now it reads as “[..] Data Management (outliers detection)”

This modification aligns the diagram with the methodology described in the manuscript, ensuring consistency between the figure and the text

Comment 11: Page 4, line 123: Since the present study is observational and not experimental, the heading of this section could be changed to “Procedures” or “Testing Procedures”.

Response 11: Thank you for your practical comment. We acknowledge that the current heading may suggest an experimental approach, whereas this study is observational. To ensure methodological accuracy, we have revised the heading on Page 4, line 141, changing it to “Testing Procedures”, which better reflects the nature of the study.

This modification aligns with the study design and avoids potential misinterpretations.

Comment 12: Page 4, line 127: It’s not very intuitive what “with phonation adapted to maintain this distance” mean – did the distance automatically change based on the speech response?

Response 12: Thank you for your comment. We acknowledge that the phrase “with phonation adapted to maintain this distance” may not be entirely clear. The intended meaning is that participants were instructed to maintain a constant distance from the microphone during phonation to ensure consistency in voice recordings, rather than the distance changing automatically in response to speech.

To clarify this, we have revised the text on Page 4, line 145- 147, to explicitly state that the distance was controlled by participant instruction: “[…] participants were instructed to maintain a constant distance from the microphone during phonation to ensure consistency in voice recordings.”

Comment 13: Page 4, Table 1: If possible, it could be helpful to describe each parameter separately (e.g., what’s the difference between Pr2 and Pr3?)

Response 13: Thank you for your constructive comment. We acknowledge that describing each parameter separately would improve clarity. To address this, we have updated Pages 5 and 6 of Table 1 by providing brief individual descriptions for each biomechanical parameter and adding a clearer distinction between similar parameters (e.g., Pr2 and Pr3).

Additionally, the Clinical Relevance column has been reviewed to ensure that each parameter's functional implications are well-defined. These modifications enhance the interpretability of the table and its relevance to voice assessment in ALS.

Parameter

Description

Clinical relevance

Pr1

Fundamental frequency. Normal range: 180- 240Hz (females)/ 105- 139Hz (males)

Indicator of pitch control and vocal fold vibration stability

Pr2

Ratio of cycles in the closing phase between free edges

Indicates regularity in vocal fold closure

Pr3

Increased open phase and asymmetry in free edge vibration

Indicates irregularity in the dynamics of the free edge during the open phase

Pr4

Percentage of time spent in the approximation phase for glottal closure

Affects vocal efficiency and closure coordination

Pr5

Percentage of time vocal folds remain separated

May indicate hypofunctional phonation

Pr6

Duration of the opening phase (vocal fold separation)

Associated with airflow dynamics during phonation

Pr7

Duration of the opening phase (vocal fold approximation)

Reflects closure speed and coordination

Pr8

Tension associated with glottal closure (hyperfunctional glottis)

Suggests excessive muscular tension affecting phonation

Pr9

Glottal closure force

May indicate compensatory mechanisms or phonatory inefficiency

Pr10

Index of optimal energy use during voice production

Evaluates phonatory efficiency and voice sustainability

Pr11

Incomplete glottal closure (gap between edges)

May contribute to breathy voice quality or phonatory inefficiency

Pr12

Alteration in the proportion of the closing phase (gap size)

Can reflect structural or functional glottal insufficiency

Pr13

Instability in maintaining vibratory tension within a single vocal cycle

Related to muscular instability, due to vocal fatigue or neurological alteration

Pr14

Instability in maintaining vibratory tension within a single vocal cycle

It correlates with instability in maintaining volume, related to intentional voice tremor, due to fatigue or neurological

Pr15

Degree of vibratory blockages throughout the voice sample

May indicate phonatory spasms or irregular vocal fold movement.  Correlates with spasmodic dysphonia

Pr16

Separation between the edges of the vocal folds

It correlates with an explosive vocal attack

Pr17

Mucosal wave correlate observed during the closing phase

Related to the laxity or viscosity of the free edge during closure

Pr18

Mucosal wave correlate observed during the opening phase

Related to the laxity or viscosity of the free edge during opening

Pr19

Edema correlate observed during the closing phase

Suggests possible inflammation or structural changes

Pr20

Edema correlate observed during the opening phase

Suggest edematous injury or excessive laxity at the free edge during closing vibration

Pr21

Structural imbalance index

It correlates with slight alterations in the structure of the free edge

Pr22

Correlates with potential structural alterations in the free edge mass or supraglottic structures

It correlates with a stable element (mass) that stands in the way of glottic closure

Comments 14: Page 5, Table 2: Since F0 appeared both in Table 1 (biomechanical parameters) and Table 2 (acoustic parameters), I was wondering whether F0 was included once or twice in the subsequent statistical models? From the figures it seems that both F0 and Pr1 were included in the models, which would be a duplicate of the same parameter.

Response 14: Thank you very much for your insightful comment. While F0 (Table 2) and Pr1 (Table 1) both represent fundamental frequencies, they were measured using different tools: F0 was obtained via acoustic analysis with Praat, whereas Pr1 was derived from biomechanical analysis with Voice Clinical Systems®.

Although their normative values are the same, slight differences may arise due to the measurement method. To clarify this distinction, we have updated the description of F0 in Page 6 Table 2 to explicitly state this.

Parameter

Description

Normal Range

Fundamental frequency

As in biomechanical analysis

Fundamental frequency (acoustic analysis, Praat). Same normative values as Pr1 but may slightly differ due to measurement method

180- 240Hz (females)/ 105- 139Hz (males)

Shimmer

Amplitude perturbation

0-      3%

Jitter

Fundamental frequency variation between cycles

~1%

Harmonics to Noise Ratio (HNR)

Signal to noise measure

Values < 20 indicate dysphonia

Comment 15: Page 5, Table 3: Were the GRABS scale parameters used for the subsequent analysis? It doesn’t seem to be included in the figure or the descriptions. If GRABS data were collected but not used in the analysis, please include a justification for this choice.

Response 15: Thank you for your valuable comment. The GRABS scale parameters were included in the analysis; however, no significant differences were found between groups. As a result, these variables were not emphasized in the figures or main results.

To ensure clarity, we have added a statement in the Results section (Page 10, Lines 299- 301) explicitly mentioning that GRABS scale parameters were analyzed but did not show statistically significant differences. Now it reads as: “On the other hands, no significant differences were found in GRABS scores between ALS and non-ALS groups, indicating that perceptual voice evaluation alone may not be sensitive enough to differentiate between these populations.”

Comment 16: Page 6, line 160: Korrigan Toolbox is a less known stats tool compared to SPSS and R. It would be great if a citation (of a peer-reviewed paper) and/or a hyperlink to the website could be added to show the credibility of the toolbox.

Response 16: Thank you for your insightful comment. To enhance clarity and transparency in our methodology, we have revised the manuscript to explicitly mention that the Korrigan Toolbox operates within MATLAB, a well-established computational environment for advanced multivariate analysis.

Additionally, we have included references for the methodology employed.

The revised text can be found on Page 8, line 197- 202.

"Statistical analyses were conducted using Korrigan Toolbox within MATLAB (MathWorks, USA), a computational framework designed for advanced multivariate analysis. Korrigan Toolbox provides specialized statistical functions that facilitate data processing in the context of voice analysis. Further details on its functionalities can be accessed at [version 0.1, Korrigan Sciences Ltd., U.K]."

Comment 17: Page 6, line 163: Please spell out the full name of OPLS here since it’s the first appearance and it’s not spelled out anywhere else in the main body of the paper.

Response 17: Thank you for your comment. We acknowledge that OPLS was introduced without spelling out its full name. To improve clarity, we have now revised Page 8, line 206, to explicitly define as Orthogonal Projections to Latent Structures Discriminant Analysis (OPLS- DA) at its first appearance in the manuscript.

This modification ensures consistency and readability for all readers.

Comment 18: Page 6, line 164: OPLS is usually used for categorical outcomes, but Barthel Index and ALSFRS-R are continuous outcome variables. It would be very helpful if a few sentences could be added here to explain how OPLS was performed on the continuous variables. For example, did the authors transform the Barthel Index and ALLSFRS-R into categorical variables by dividing the patients into a high functionality group and a low functionality group?

Response 18: Thank you very much for your valuable comment regarding the use of OPLS-DA with continuous outcome variables such as the Barthel Index and ALSFRS-R. We appreciate the opportunity to clarify this aspect of our methodology.

In our study, we utilized the Korrigan Toolbox in MATLAB, which allows OPLS-DA to handle continuous Y variables. This approach enables us to retain the full variability of functional scores and establish more precise relationships between voice parameters and ALS functional status, avoiding the need for arbitrary categorization.

To ensure clarity for readers, we have now explicitly stated this methodological detail in Section 2.4 Data Analysis of the revised manuscript Pages 8, Lines 222- 231. The following amendment has been included:

"In this study, the OPLS-DA analysis was performed using the Barthel Index and ALSFRS-R as continuous outcome variables. While OPLS-DA is traditionally applied to categorical outcomes, the Korrigan Toolbox in MATLAB enables the use of continuous Y variables. This approach preserves the full variability of the functional scores, allowing for a more nuanced assessment of their relationship with voice parameters without the need for arbitrary categorization. By applying this method, the analysis captures subtle associations between biomechanical and acoustic voice measures and ALS functional status, ensuring a comprehensive statistical evaluation."

Comment 19: Also, from the results section, it seems that the patient groups (ALS vs. Control, Bulbar ALS vs. Control, Spinal ALS vs. Control, and Bulbar vs. Spinal) also serve as the outcome Y variable in OPLS.

Response 19: We appreciate your observation regarding the use of patient groups as the outcome Y variable in the OPLS analysis. Indeed, this classification approach aligns with the standard methodology in supervised multivariate analysis, where categorical variables (Y) are modeled against predictor variables (X, the matrix of independent biomechanical and acoustic voice parameters) to identify relevant patterns of differentiation.

In our study, the Y variable corresponds to the categorical patient groups, while the X variables include the biomechanical and acoustic voice parameters. This design allows for the identification of voice features that best distinguish between the studied groups. As depicted in Figure 2, Page 10 the Y-axis represents the patient classification, whereas the X-axis shows the distribution of voice parameters.

To enhance clarity, we have now explicitly stated in the Methods section (Page 8, Line 220-222) that the OPLS model was structured with group classification as the Y variable and voice parameters as the X variables:

“[…] and the categorical patient groups (ALS vs. Control, Bulbar ALS vs. Control, Spinal ALS vs. Control, and Bulbar vs. Spinal) as dependent variables (Y).”

Comment 20: Page 6, line 174: It could be helpful to add a sentence to justify why including smoking habit in the analysis. Specifically, to what extent is smoking relevant to voice features, voice disorders, and ALS?

Response 20: We appreciate your observation regarding the role of smoking habits in the analysis. Tobacco use is known to affect vocal function, leading to structural changes such as vocal fold edema and altered vibratory dynamics. To assess whether smoking habits influenced voice features, we conducted a separate model to evaluate the parameters affected by smoking. Our analysis showed that smoking was not associated with ALS-related voice parameters. Therefore, smoking was not included as a covariate in the final model. We have now clarified this in the Methods section. Page 7, Lines 189- 194.

"Smoking habit was initially considered due to its effects on vocal function, such as inducing structural and functional alterations in the vocal folds, including edema and stiffness, which can impact voice parameters. However, after running a separate model to assess which parameters were influenced by smoking, we found no association between smoking and ALS-related voice features. Therefore, smoking was not included as a covariate in the final analysis."

Comment 21: Page 6, line 177: Sex is a categorical data and therefore cannot be described as “normally distributed” or tested for normal distribution. Might worth considering replacing it with a chi-square test if the authors would like to check whether the proportions of female and male patients are significantly deviant from 50%.

Response 21: Thank you for pointing this out. The reviewer is right, sex is a categorical variable and cannot be analyzed in terms of normality as a numeric variable. We have revised our statistical approach and now use a “chi-square test to assess whether the distribution of male and female participants deviates significantly from an expected 50% distribution” Page 9, Lines 246- 247.

Additionally, sex was investigated in our analysis, as previous research has shown physiological differences in voice production between males and females. These modifications are now explicitly described in the Methods section under the new subsection on covariates Page 7, Lines 184- 188.

“We investigated what voice parameters were influenced by sex, and how voice parameters were modified within each sex. The results showed that sex was only associated with Fo and Pr1, which are voice parameters not relevant for ALS patients. This approach ensured that the observed differences in voice parameters were primarily attributable to ALS-related changes rather than inherent differences between male and female speakers.”

Comment 22: Page 7, lines 189-191: How was the lack of significant differences determined? Specifically, was it based on the OPLS-DA analysis, or t-tests? It could be helpful to include the stats values here to support the lack of significant differences.

Response 22: Thank you for your valuable comment regarding the reporting of statistical values for non-significant comparisons in Section 3.2.1. To clarify, the OPLS-DA model indicated differences in several vocal parameters; however, given its relatively low explanatory power (R² = 0.16 for bulbar ALS vs. controls and R² = 0.07 for spinal ALS vs. controls), we performed additional statistical validation using independent t-tests to confirm the significance of individual parameters. While Pr12 showed a statistically significant difference in bulbar ALS patients (p = 3.08e-05), other comparisons did not reach statistical significance. In the revised manuscript, we have now explicitly stated that the remaining comparisons were not statistically significant, as non-significant values are not typically presented in detail. Please, find below the amended version Page 9, Lines 266- 278:

“The OPLS-DA models comparing bulbar ALS patients (R² = 0.16) and spinal ALS patients (R² = 0.07) to the control group were relatively weak, possibly due to the small sample size (Figure 2). Given the limited explanatory power of these models, additional statistical validation was performed using independent t-tests to assess the significance of individual parameters previously selected by OPLS-DA While the analysis suggested changes in several Pr signals for bulbar ALS, only Pr12 showed a statistically significant difference in Pr12 between the groups (t(56.57) = 4.53, p = 3.08e-05), with the application of the False Discovery Rate (FDR) correction to account for multiple comparisons. The 95% confidence interval ranged from 10.25 to 26.47. The mean Pr12 value was 21.27 for group 0 and 2.91 for group 1 (bulbar). For spinal ALS, Pr21 did not reach statistical significance in the t-test (p > 0.05). Similarly, all other comparisons failed to achieve statistical significance, confirming that only Pr12 exhibited a robust difference in bulbar ALS patients.”

Comment 23: Page 7, lines 193-195: What’s the criteria of parameter selection – What’s the cut-off R^2 value? It’s described as “Warm colors” in the caption of Figure 2 but that’s vague. Since there are R2Y and R2X values in OPLS, it could also be helpful to add a sentence to describe what R2 values here means specifically.

Response 23: Thank you very much for your comment regarding the criteria for parameter selection in OPLS-DA, we have now clarified the analysis in the revised manuscript. We acknowledge that, unlike p-values in traditional statistical methods, OPLS-DA models do not have a universally accepted fixed cutoff for the R² value. As such, we did not define a specific R² cutoff in the original manuscript. Instead, we emphasize the use of the Q²Y value to evaluate the goodness of prediction, where a positive Q²Y value indicates a reliable model. Furthermore, we inspect the models for overfitting by analyzing the scatter of the model scores (T) versus the cross-validated scores (Tcv). If the model deviates significantly from the diagonal, indicating potential overfitting, it is discarded. This approach reflects a more flexible strategy in evaluating the performance of OPLS-DA models, which is more aligned with the nature of multivariate modeling, where interpretability and predictive accuracy take precedence over rigid thresholds. We have also included further details on the model validation process, which involved seven-fold cross-validation to assess predictive accuracy. We carefully inspected the models to ensure that no overfitting occurred. Additionally, the model loadings plot was color-coded to highlight features most strongly associated with the discriminant components, with warm colors (closer to red) indicating the most discriminative features and cold colors (closer to blue) indicating less significant features.

We hope these clarifications help explain the model evaluation process more thoroughly. We appreciate your input, as it has helped refine the manuscript. The new section in Page 8, Lines 203- 216 reads as follows: 

“[…] Matrix normalization was carried out using a median-based probabilistic quotient method (Dieterle et al., 2006). The analysis of the samples followed a two-step approach. Initially, unsupervised Principal Component Analysis (PCA) was used for quality control and to assess the data structure. This was followed by a supervised pairwise Orthogonal Projection to Latent Structures Discriminant Analysis (O-PLS-DA) (Bylesjö et al., 2007; Cloarec et al., 2005), which allowed identification of specific modulations driven by the factor of interest. The O-PLS-DA models were validated using seven-fold cross-validation, and their performance was evaluated based on the Q²Y value (goodness of prediction). The models were carefully inspected to check for overfitting, with particular attention to the scatter of the model scores (T) and cross-validated scores (Tcv), where overfitted models would deviate from the diagonal and be discarded. Additionally, the model loadings plot was color-coded to highlight features most strongly associated with the discriminant component in warm colors (closer to red), while those in cold colors (closer to blue) were not considered discriminant.”

Comment 24: Page 7, line 198: Were the results of t-tests corrected for multiple comparisons (by using FDR or Bonferroni correction etc)? Also please include t value and DF when reporting the result of t-tests.

Response 24: Thank you for your valuable feedback. In response to your request, we have revised the manuscript to explicitly include the t-value (t = 4.53), degrees of freedom (DF = 56.57), and p-value (p = 3.08e-05). Additionally, we confirm that the False Discovery Rate (FDR) correction was applied to control for multiple comparisons.

A Welch Two-Sample t-test was conducted to compare the variables previously selected by O-PLS DA between the two ALS subtypes (bulbar vs. spinal ALS). The results revealed a significant difference in Pr12 between the groups (t (56.57) = 4.53, p = 3.08e-05), even after FDR correction. The 95% confidence interval ranged from 10.25 to 26.47, with a mean Pr12 value of 21.27 for group 0 and 2.91 for group 1 (bulbar ALS).

To ensure transparency and clarity, we have explicitly incorporated this information into the manuscript.

The following amendment has been included in Page 9, Lines 270- 275:

"[…] While the analysis suggested changes in several Pr signals for bulbar ALS, only Pr12 showed a statistically significant difference between the groups (t (56.57) = 4.53, p = 3.08e-05), with the application of the False Discovery Rate (FDR) correction to account for multiple comparisons. The 95% confidence interval ranged from 10.25 to 26.47. The mean Pr12 value was 21.27 for group 0 and 2.91 for group 1 (bulbar ALS)."

These findings indicate a robust difference between the groups, reinforcing the validity of our statistical approach.

Comments 25: Page 7, Figure 2: What is the unit of the vertical axis? The scale of the vertical axis is different between panels a) and b), making it confusing. This applies to the other figures as well, especially figures 5 & 6 where the range of the vertical axis differs drastically from the other figures.

Response 25: Thank you for your constructive comments. Regarding your observation about the consistency of the y-axis units and scale across figures (particularly Figures 2, 5, and 6), we would like to clarify that the y-axis scale cannot be consistent between figures, as they represent different OPLS-DA models. Each model has its own set of latent components and results, which reflect the variability in the data based on the predictors used and the nature of the dependent variable (Y). In OPLS-DA models, the y-axis scale is adjusted according to the magnitude of the model’s output for each specific analysis. This is because the scores (T) and cross-validated scores (Tcv) are directly influenced by the dispersion and underlying structure of the data in each model, which may vary from one model to another. Normalizing across different models is not appropriate in this context, as the latent components can have different variances and scales. Instead of normalizing the y-axis scales between figures, each model should be represented with its own scale to accurately reflect the relationships in the data and the results obtained from each analysis. To avoid confusion, we have ensured that each figure includes a clear legend, and the description of the OPLS-DA models and their respective axes is detailed in the manuscript, so readers can properly interpret the variations in values across figures. We hope this explanation clarifies why the y-axis scale is not consistent across figures, and we appreciate your understanding.

Comment 26: The vertical directions of panels a) and d) are swapped – it might be more straightforward if the downward arrow corresponds to the control group in both panels.

Response 26: Thank you for your insightful comments. Regarding your suggestion to standardize the "vertical directions" in Figure 2, we believe this refers to the loadings in the OPLS-DA models.

In OPLS-DA, the loadings represent the weights or contributions of each variable to the discriminant components of the model. The direction of the loadings (vertical or otherwise) reflects the contribution of each feature (e.g., voice parameters) in the model and how they relate to the variation in the data that is being explained by the latent components. These loadings are typically plotted on the y-axis of loadings plots, and their magnitude and direction (upward or downward on the axis) indicate the strength and direction of association with the component.

It's important to note that the vertical direction of the loadings (positive or negative) in OPLS-DA is inherently tied to the specific component being analyzed. Therefore, the vertical direction cannot be standardized across figures, as each OPLS-DA model is unique, and the loadings will naturally vary depending on the data and the components selected for each analysis. The lack of a standardized "vertical direction" is a reflection of the model’s flexibility to adapt to the specific patterns and variances in the data, rather than a flaw in the analysis. In addition, whether or not the control group plots on the positive or negative Y axis region is not a decision of the researcher, but a result of the model itself.

To aid clarity, we have ensured that the loadings plots are properly labeled with explanations about what each axis represents and have added an explanatory sentence in the figure legend (Page 10, Lines 287- 289), explicitly stating that:

"Loadings (bar plots) represent the contribution of each variable to the discriminant components and may vary across models depending on the data and components analyzed."

We hope this explanation provides further clarity regarding the representation of the loadings and why their direction may differ across panels. Thank you for your valuable feedback.

Comment 27: The font size in panel c) needs to be increased.

Response 27: Thank you for pointing out the readability issue in panel (c) of Figure 2. While the figure legend provides an explanation of the violin plot and statistical results, we acknowledge that the font size within the plot itself may have been too small for optimal readability.

To address this concern, we have increased the font size of the labels and statistical values in panel (c) to ensure greater clarity. This modification has been incorporated into the revised Figure 2 in Page 10.

We appreciate your feedback and believe this change enhances the figure's readability.

Comment 28: Page 8, lines 238-239: Based on this paragraph, it seems that the relationship between Barthel Index and voice analysis was only assessed in ALS patients. However, Figure 4 suggests both ALS and the control group were included in the analysis. I was wondering which one is the accurate description of the analysis. If both groups were included, was the Barthel Index treated as the only outcome variable (regardless of whether the patients are from the ALS or control group), or was the “ALS vs. control” included somewhere in the analysis as a predictor/covariate?

Response 28: Thank you for your question regarding the role of the Barthel Index in our analysis. In this study, the Barthel Index was used as a dependent variable (outcome measure) to assess whether both acoustic and biomechanical voice parameters could predict the functional status of ALS patients.

Since the Barthel Index is designed to evaluate functional independence, the analysis aimed to determine whether specific voice alterations correlate with lower functional scores in ALS patients. The ALS vs. control classification was not included as a predictor or covariate in this model, as the Barthel Index was not directly assessed in controls. Instead, non-ALS participants were assigned the maximum Barthel score to facilitate comparisons.

To ensure clarity, we have revised the Methods section (Page 7, Lines 175- 178) to explicitly state that the Barthel Index was analyzed as a dependent variable in this predictive analysis:

“The Barthel Index was used as a dependent variable to evaluate whether acoustic and biomechanical voice parameters could predict functional status in ALS patients. Control individuals, being voice patients without neurodegenerative disorders, were assumed to have full functional independence and thus assigned the maximum Barthel Index score.”

We appreciate your valuable feedback, which has helped us refine the methodological description of our study.

Comment 29: Page 8, line 239: “the elevated value of Pr15” – this doesn’t seem to be consistent with Figure 4. The bar for Pr15 is barely visible even after zooming in. (To clarify, the current figure is not necessarily wrong, since it could be that Pr15 is comparable between ALS and control group but showed a significant correlation with Barthel Index only in the ALS group. It’s just not a very effective way to illustrate the results describe in the text.)

Response 29: Thank you for highlighting this point. As noted, the magnitude of the bar for Pr15 in Figure 4 is small, which could visually suggest minimal differences between groups. However, the color coding of bars (reflecting the R² correlation values) is the critical element to interpret statistical significance in these OPLS-DA loading plots. The legend already clarifies this, indicating that significant correlations are represented by warmer colors, independent of bar size.  Additionally, we have replaced the wording "the elevated value of Pr15" with " the significant value of Pr15 " to avoid confusion in Page 12, Line 350. This adjustment ensures clearer interpretation, complementing the figure legend explanation.

Comment 30: Page 9, Figure 4: It might be helpful to modify the figure or add a panel to show which parameters are positively and negatively corelated with Barthel Index, in the same format as the current panel. This also applies to ALSFERS-R scores in Figure 5.

Response 30: Thank you for your suggestion. We agree that clearly illustrating which parameters correlate positively or negatively with clinical measures (Barthel Index and ALSFRS-R) is essential. Figure 4 already includes a detailed clarification explaining how to interpret the direction (positive or negative) of bars. To enhance consistency and further clarify Figure 5, we have now included a similar brief explanatory note, specifying explicitly that bar directions and magnitudes reflect associations exclusively with ALSFRS-R scores and should be interpreted independently of other figures.

The modified text on Page 14, lines 378- 379 now reads: "Note: Bar directions and magnitudes specifically reflect associations with ALSFRS-R scores and should be interpreted independently from other figures (e.g., Figure 4)."

We appreciate your valuable feedback, which helps improve the clarity of our manuscript.

Comment 31: I was also confused by the difference in the height and directions of the bars in Figures 4 and 5. If both figures are showing the differences in vocal parameters between ALS and control groups, as described in the figure captions, they should have the same height and direction in Figures 4 and 5, only different colors representing different R2 values. However, currently they are very different especially for Pr12-21.

Response 31: Thank you for highlighting this point. We understand your concern. To clarify, Figures 4 and 5 depict independent analyses examining distinct clinical outcomes: Figure 4 explores associations with the Barthel Index (functional status), whereas Figure 5 addresses relationships with the ALSFRS-R scale (disease severity). Thus, differences in bar direction and magnitude between these two figures are expected, as each reflects unique statistical relationships with separate clinical measures. Each figure legend independently describes the corresponding clinical variable analyzed, so no direct comparison between these two figures was intended.

We appreciate your comment, which helped clarify this methodological distinction.

Comment 32: Page 10, Figure 6: Were the sex and smoking analyses performed on all patients, or ALS patients only?

Response 32: Thank you for your question. To clarify, the analyses presented in Figure 6 regarding the influence of sex and smoking habits were performed only on ALS patients. These variables were evaluated specifically within the ALS patient group to assess their potential impact on vocal parameters without introducing variability from the control group.

We have clarified this explicitly in Page 15, Lines 398 and 403 in the figure legend (Figure 6) to avoid further confusion, adding “[..] in ALS patients” and changing “Sex 2” for “women”.

Thank you for helping us enhance the clarity of the manuscript.

Comment 33: Also, it is unclear which smoking habit “F0/F1” refers to, and which sex “Sexo1/Sexo2” refers to. Please replace the labels in the legend with the actual description of the group e.g., “Smoker/Non-smoker”.

Response 33: Thank you for highlighting this ambiguity. We agree that the original labels were unclear. To address this, we have updated the labels in Figure 7 (in Page 15) to clearly indicate the groups being compared. Specifically:

"F0" and "F1" have been replaced by "Non-smoker" and "Smoker", respectively.

"Sexo1" and "Sexo2" have been replaced by "Male" and "Female", respectively.

This adjustment improves clarity, making the figure easier to interpret.

We appreciate your valuable feedback.

Comment 34: Page 11, line 327: Pr9 and Pr12 also showed differences, just in the opposite direction.

Response 34: Thank you for your observation. The directional differences observed for Pr9 and Pr12 were indeed noted and discussed explicitly in the results section, including detailed statistical validation for Pr12 (t (56.57) = 4.53, p = 3.08e-05, FDR-corrected). Thus, we have chosen not to repeat this information extensively in the discussion, maintaining focus on the clinical interpretation of the most robust findings. However, if you consider necessary further emphasis in the discussion, we can briefly reference the detailed results provided previously.

Comment 35: Page 12, line 344: Pr5 and Pr18 showed increase in the bulbar group, while Pr9 showed decrease in the bulbar group, compared to the spinal group.

Response 35: Thank you for highlighting this aspect. Upon reviewing the discussion carefully, we identified the confusion you mentioned regarding parameters Pr5, Pr9, and Pr18. Indeed, as you correctly observed, Pr5 and Pr18 were increased in the bulbar ALS group, reflecting changes in rhythm and prosody. However, Pr9, which represents the effort required to achieve and maintain maximum glottal closure, was specifically increased in the spinal ALS group, reflecting a compensatory hyperfunctional behavior.

To clarify and avoid confusion, we have adjusted the text (Page 16, Lines 460- 464 in Discussion) to explicitly reflect that Pr9 was increased in the spinal ALS group relative to the bulbar ALS group, rather than indicating an increase across all three parameters in the same direction.

Now it reads as "When comparing bulbar and spinal ALS patient groups, significant differences in vocal parameters were observed, suggesting distinct biomechanical adaptations in each phenotype." instead of “In contrast, when comparing the bulbar and spinal ALS patient groups, an increase in the parameters P5, Pr9, and Pr18 was observed, suggesting that the two types of ALS affect vocal production differently.”

Thank you for helping us enhance the clarity and accuracy of this important distinction in our discussion.

Reviewer 2 Report

Comments and Suggestions for Authors

I like the fact that authors present some preliminary results,, but clarify that due to the small sample size and the cross-sectional nature of the study, their findings should be confirmed in future trials. 

Major changes:

- Providing the sample size calculation, without explaining which actual statistic - research question drove your calculations may be misleading. Please, add a separate section in Methods that explain your hypotheses or research questions, and then relate these to your statistical analysis (as described in 2.4. Data Analysis). Then it will be clear why each stat / PCA was selected.

- In continuation with the above, has you study been pre-registered? If yes, please present the analyses that were held outside your initial protocol as "post-hoc". 

- Please, discuss in the discussion how and why "voice analysis represents a promising, non-invasive, and objective tool 420 for evaluating speech motor deficits in ALS" as biomarker. When and how would it be used to improve prognosis (which is definitive) or quality of life? In brief, decrease the size of the discussion that repeats the results, and focus on what is the research and clinical impact you envisage.

- Why was Barthel and ALSFRS-R compared in ALS and non-ALS controls (as mentioned in p.9, line 252)? What was the clinical and statistical motivation of using this scale on non-neurological patients?

Minor changes:

-Please clarify in the introduction the difference between motor neuron disease and ALS.

- Table 5. Please provide the control group parameters, too.

Author Response

Response to Reviewer 2 Comments

Dear reviewer,

We appreciate the time and dedication you have put into reviewing our manuscript and we will now respond to your comments point by point, which will undoubtedly improve its quality.

Comment 1: I like the fact that authors present some preliminary results but clarify that due to the small sample size and the cross-sectional nature of the study, their findings should be confirmed in future trials. 

Response 1: We sincerely appreciate the reviewer's positive feedback. We believe it is essential to transparently communicate the limitations of our study, including its small sample size and cross-sectional design, to ensure that our findings are interpreted appropriately. By acknowledging these constraints, we aim to emphasize the need for future longitudinal studies with larger cohorts to validate our preliminary results and further explore the clinical applications of voice analysis in ALS monitoring.

Additionally, we would like to mention that we have continued recruiting patients to expand our dataset for future analyses. However, given that ALS is a rare disease, the sample size remains limited despite our ongoing efforts. We remain committed to strengthening the robustness of our findings as data collection progresses.

Comment 2: Providing the sample size calculation, without explaining which actual statistic - research question drove your calculations may be misleading. Please, add a separate section in Methods that explain your hypotheses or research questions, and then relate these to your statistical analysis (as described in 2.4. Data Analysis). Then it will be clear why each stat / PCA was selected.

Response 2: Thank you for your valuable suggestion. We agree that explicitly outlining the research questions and hypotheses before presenting the statistical analysis can enhance the clarity of our methodological approach.

In response to a similar comment from Reviewer 1, we have now clarified the rationale behind our sample size calculation by explicitly defining the prevalence rates used in our estimation. Additionally, we have ensured that the research questions guiding our statistical approach are clearly stated within the Methods section. These modifications should make the connection between our hypotheses and the statistical methods used (PCA, OPLS-DA) more transparent.

  • The sample size calculation has been clarified, specifying the prevalence rates used for estimating group differences. The proportions refer to the estimated prevalence of voice disorders in ALS patients (80%) and in the general population without neurological disorders (10%).

The 80% prevalence in ALS patients is supported by evidence indicating that dysphonia is a frequent symptom in ALS, especially in bulbar-onset cases where voice impairments are among the earliest manifestations. According to the Regional Association of People Affected by Amyotrophic Lateral Sclerosis (Asociación Regional de Afectados de Esclerosis Lateral Amiotrófica, ARAELA), voice alterations are a common feature in ALS progression (ARAELA).

The 10% prevalence of voice disorders in the general population is within the range reported in previous epidemiological studies. Some studies estimate that 1%–17% of the general population has experienced voice disorders (Murtró, 2019; Lyberg-Ahlander, 2018), while others indicate that the prevalence can be higher in specific subpopulations, such as teachers, where up to 20% present voice disorders (Hernández, 2020).

Additionally, according to the Spanish Society of Otolaryngology and Head and Neck Surgery (SEORL-CCC, 2017), approximately 7.7% of the general population (equivalent to 1 in 13 people) experience voice disorders, yet most cases go untreated.

Based on this evidence, a 10% baseline prevalence was considered a reasonable approximation for the general non-neurological population.

To clarify this point, we have revised the Methods section (Page 3, lines 119-126) to explicitly define that the two proportions correspond to the prevalence of voice disorders in ALS patients and in the general population, along with references supporting these estimations.

“[…] 0.8 (estimated prevalence of voice disorders in ALS patients) and 0.1 (estimated prevalence of voice disorders in the general population). These values were selected according to previously reported prevalence rates of voice alterations in ALS patients (~80%) and general populations (~1–17%) (Murtró, 2019; Lyberg-Ahlander, 2018) with the Spanish Society of Otolaryngology (SEORL-CCC, 2017) estimating a 7.7% prevalence in the general population. A 10% baseline prevalence was thus considered a reasonable approximation for the general population”

  • We have added a new section in Methods explicitly outlining the research hypotheses and how they relate to our statistical choices. To enhance clarity, we have now explicitly stated in the Methods section (Page 8, Line 220-222) that the OPLS model was structured with group classification as the Y variable and voice parameters as the X variables:

“[…] and the categorical patient groups (ALS vs. Control, Bulbar ALS vs. Control, Spinal ALS vs. Control, and Bulbar vs. Spinal) as dependent variables (Y).”

  • The rationale behind applying PCA for data exploration and O-PLS-DA for classification analysis: we have now clarified the analysis in the revised manuscript. We acknowledge that, unlike p-values in traditional statistical methods, OPLS-DA models do not have a universally accepted fixed cutoff for the R² value. As such, we did not define a specific R² cutoff in the original manuscript. Instead, we emphasize the use of the Q²Y value to evaluate the goodness of prediction, where a positive Q²Y value indicates a reliable model. Furthermore, we inspect the models for overfitting by analyzing the scatter of the model scores (T) versus the cross-validated scores (Tcv). If the model deviates significantly from the diagonal, indicating potential overfitting, it is discarded. This approach reflects a more flexible strategy in evaluating the performance of OPLS-DA models, which is more aligned with the nature of multivariate modeling, where interpretability and predictive accuracy take precedence over rigid thresholds. We have also included further details on the model validation process, which involved seven-fold cross-validation to assess predictive accuracy. We carefully inspected the models to ensure that no overfitting occurred. Additionally, the model loadings plot was color-coded to highlight features most strongly associated with the discriminant components, with warm colors (closer to red) indicating the most discriminative features and cold colors (closer to blue) indicating less significant features.

We hope these clarifications help explain the model evaluation process more thoroughly. We appreciate your input, as it has helped refine the manuscript. The new section in Page 8, Lines 203- 216 reads as follows: 

“[…] Matrix normalization was carried out using a median-based probabilistic quotient method (Dieterle et al., 2006). The analysis of the samples followed a two-step approach. Initially, unsupervised Principal Component Analysis (PCA) was used for quality control and to assess the data structure. This was followed by a supervised pairwise Orthogonal Projection to Latent Structures Discriminant Analysis (O-PLS-DA) (Bylesjö et al., 2007; Cloarec et al., 2005), which allowed identification of specific modulations driven by the factor of interest. The O-PLS-DA models were validated using seven-fold cross-validation, and their performance was evaluated based on the Q²Y value (goodness of prediction). The models were carefully inspected to check for overfitting, with particular attention to the scatter of the model scores (T) and cross-validated scores (Tcv), where overfitted models would deviate from the diagonal and be discarded. Additionally, the model loadings plot was color-coded to highlight features most strongly associated with the discriminant component in warm colors (closer to red), while those in cold colors (closer to blue) were not considered discriminant.”

We appreciate your insightful suggestion, which has helped us further refine the clarity of our manuscript.

Comment 3:  In continuation with the above, has you study been pre-registered? If yes, please present the analyses that were held outside your initial protocol as "post-hoc". 

Response 3: Thank you for your insightful question. This study was not pre-registered as it was designed as an exploratory observational study to investigate biomechanical and acoustic voice parameters in ALS patients. The analyses were planned to assess differences between groups and their relationship with functional scales, as outlined in the Methods section. However, based on reviewers’ suggestions, additional statistical analyses, such as the ROC curve analysis, were performed to further explore the discriminative ability of voice parameters. These additional analyses are now described as post-hoc analyses in the revised manuscript.

Additionally, the following modifications were made to the manuscript:

Methods Section (Page 8, Lines 232-236): "To assess the sensitivity and specificity of voice parameters in differentiating ALS subtypes, we performed a classical ROC curve analysis for individual biomarkers and an overall classification analysis using an orthogonal partial least squares discriminant analysis (O-PLS-DA) model. The model was trained with one latent variable and features selected based on AUROC ranking."

Results Section (Page 11, Lines 323-329): "The ROC analysis demonstrated that individual voice parameters, such as Pr18 (AUC: 0.717), Pr9 (AUC: 0.688), and Pr5 (AUC: 0.682), showed moderate discriminative ability for ALS subtype differentiation (panels a–f). The overall O-PLS-DA model incorporating these features achieved an AUC of 0.71 (95% CI: 0.452–0.895), with sensitivity and specificity values varying across cross-validation iterations (panel m). The confusion matrix (panel n) and prediction accuracy distribution (panel o) further illustrate model performance."

Comment 4: Please, discuss in the discussion how and why "voice analysis represents a promising, non-invasive, and objective tool 420 for evaluating speech motor deficits in ALS" as biomarker. When and how would it be used to improve prognosis (which is definitive) or quality of life? In brief, decrease the size of the discussion that repeats the results, and focus on what is the research and clinical impact you envisage.

Response 4: Thank you for your valuable suggestion. We have revised the Discussion to reduce redundancy and focus on the clinical and research implications of voice analysis as a non-invasive and objective biomarker for ALS.

To address this: We condensed sections that reiterated the results without adding new insights. Page 16, Lines 460- 464 substituting “In contrast, when comparing the bulbar and spinal ALS patient groups, an increase in the parameters P5, Pr9, and Pr18 was observed, suggesting that the two types of ALS affect vocal production differently. “ for “When comparing bulbar and spinal ALS patient groups, significant differences in vocal parameters were observed, suggesting distinct biomechanical adaptations in each phenotype.” and removing redundant text from results in lines 471- 473.

We expanded on how biomechanical voice parameters could be integrated into ALS prognosis and patient monitoring, particularly in early detection and disease progression tracking. Now it reads:  "Voice analysis is a promising, non-invasive tool for assessing speech motor deficits in ALS, with potential applications in early diagnosis and disease monitoring. Given that voice dysfunction often appears before other clinical symptoms, tracking biomechanical voice parameters may enable earlier intervention and better disease management. Additionally, integrating voice analysis into clinical practice could provide a quantitative measure of functional decline, supporting personalized therapeutic strategies and improving patient quality of life." in Page 17, Lines 523- 529.

We appreciate your feedback, which has helped refine the Discussion to emphasize the practical and translational impact of our findings.

Comment 5: Why was Barthel and ALSFRS-R compared in ALS and non-ALS controls (as mentioned in p.9, line 252)? What was the clinical and statistical motivation of using this scale on non-neurological patients?

Response 5: Thank you for your thoughtful comment. We acknowledge the importance of clarifying the rationale behind the inclusion of the Barthel Index in the analysis. The Barthel Index was used as a dependent variable to assess whether acoustic and biomechanical voice parameters could predict functional status in ALS patients. Since controls did not have neurodegenerative disorders, they were assumed to have full functional independence and assigned the maximum Barthel Index score and the ALSRFS-R. This approach allowed for a comparative analysis without altering the structure of the dataset while maintaining consistency in evaluating functional status.

To ensure clarity, we have revised the Methods section (Page 7, Lines 175- 178) to explicitly state that the Barthel Index was analyzed as a dependent variable in this predictive analysis:

“The Barthel Index was used as a dependent variable to evaluate whether acoustic and biomechanical voice parameters could predict functional status in ALS patients. Control individuals, being voice patients without neurodegenerative disorders, were assumed to have full functional independence and thus assigned the maximum Barthel Index score.”

We appreciate your valuable feedback, which has helped us refine the methodological description of our study.

Comment 6: Please clarify in the introduction the difference between motor neuron disease and ALS.

Response 6: Thank you for your valuable suggestion. We acknowledge the importance of clarifying the distinction between motor neuron disease (MND) and amyotrophic lateral sclerosis (ALS) in the Introduction to ensure conceptual precision.

Motor neuron disease (MND) is an umbrella term that encompasses a group of neurodegenerative disorders affecting motor neurons, leading to progressive muscle weakness. ALS is the most common form of MND, but other subtypes include primary lateral sclerosis (PLS) and progressive muscular atrophy (PMA), which present with different clinical and pathological features.

To enhance clarity, we have revised the Introduction (Page 2, Lines 47- 50) by explicitly defining MND and its relationship to ALS, now reads it: "Amyotrophic Lateral Sclerosis (ALS) is the most common form of motor neuron disease (MND), a group of neurodegenerative disorders affecting motor neurons. ALS specifically involves the degeneration of upper and lower motor neurons in the cortex, brainstem, and spinal cord."

We appreciate this insightful comment, as it has allowed us to improve the conceptual framework of our study.

Comment 7: Table 5. Please provide the control group parameters, too.

Response 7: Thank you for your comment. We acknowledge the importance of including the control group parameters in Table 5, Page 9 to facilitate a comprehensive comparison between ALS patients and controls.

To address this, we have updated Table 5 by adding the corresponding control group values, allowing for a clearer interpretation of the differences across groups.

This revision enhances the completeness of the data presentation and aligns with best practices for comparative analysis.

We appreciate your feedback, which has helped us improve the clarity and robustness of our results.

Sociodemographic

Clinical Score ALS

Clinical Score non-ALS

Age

63.31 ±9.45

64.02 ±7.47

N (men/ women)

39 (24/ 15)

43 (27/ 16)

Smoking (No/ Yes)

20 (24.4%)/ 19 (23.2%)

35(81.4%)/ 8(18.6%)

Time to diagnosis ALS (months)

20.56 ±24.15

Not applicable

Type of ALS: Bulbar/ Spinal

17 (20.7%)/ 22 (26.8%)

Not applicable

ALSFRS- R

35.38 ±6.98

48 ±0

Barthel

67.85 ±20.95

100 ±0

GRABS

3.92 ±4.15

5.7 ±2.25

Round 2

Reviewer 1 Report

Comments and Suggestions for Authors

Thank you for submitting the revision and the detailed cover letter explaining the changes. Most of my comments on the previous version have been sufficiently addressed. There are still a few minor outstanding comments and suggestions, as detailed below. I recommend the manuscript for acceptance after these follow-up comments have been addressed.

Page 2, lines 52-54: This is a follow-up comment on Revision 4 – are the “global incidence” data reported here also measured at a certain time point, or across a time window (e.g., 1 year)? If I understood correctly, “incidence” data represents the newly diagnosed cases within a certain time window, so the length of the time window is important for interpreting the data.

Page 9, line 247: This is a follow-up comment on Revision 22 – the p value (p = 2.17) appears to be a typo, as p values are between 0 and 1 by definition.

Page 10, Figure 2: This is a follow-up comment on Revision 25 – the authors’ explanation clarified that the direction and values of the Y axis cannot be standardized across figures since they reflect the inherent properties of different OPLS-DA models. However, it is still unclear what the values of Y axis represent (i.e., which specific parameters of the OPLS-DA model) in the bar plots. Also, panel b) seems to be a zoomed-in version of panel a), both representing the same OPLS-DA model, but these two panels are very different in the range of Y axis.

Pages 12-13, section 3.3 and Figure 5: This is a follow-up comment on Revision 28 – thank you for providing the detailed explanation in the cover letter. It could be helpful to add in the text that even though Figure 5 plotted the comparison between ALS and control groups, the model reported in this section focused on the Barthel Index within the ALS group and did not include ALS vs. control classification as an outcome variable. This might be applicable to Figure 6 as well.

Page 13, Figure 6: This is a follow-up comment on Revision 30. In lines 375-376, it states that the direction of the bars corresponds to whether the values were higher in the ALS or the control group. However, in line 378, it states that the direction reflects the associations between the ALSFRS-R scores and the vocal parameters (and based on the cover letter, it appeared that this association was only evaluated within the ALS group). It is still not very clear whether the direction of the bars represent the comparison of the values between ALS vs. control groups, or the correlations between ALSFRS-R and voice parameters within the ALS group. If the figure is based on the model assessing ALSFRS-R scores only in the ALS group, please consider renaming the arrows on the left side of the figure as “Positively/negatively correlated with ALSFRS-R scores”. This might be applicable to Figure 5 as well.

Author Response

Response to Reviewer 1 Comments (Round 2)

Dear reviewer,

We appreciate the time and dedication you have put into reviewing our manuscript and for the recognition of our efforts to promptly address your comments. We will now respond to your comments point by point, which will undoubtedly improve its quality.

Comment 1: Page 2, lines 52-54: This is a follow-up comment on Revision 4 – are the “global incidence” data reported here also measured at a certain time point, or across a time window (e.g., 1 year)? If I understood correctly, “incidence” data represents the newly diagnosed cases within a certain time window, so the length of the time window is important for interpreting the data.

Response 1: Thank you for your comment. In the manuscript, the reported "global incidence" (1.75 cases per 100,000 inhabitants) represents the annual incidence of the disease, that is, the number of newly diagnosed cases per year in each population. This metric is expressed over a standard period of one year to allow comparisons between different studies and regions.

For greater clarity, we have modified the text in the manuscript to specify that the global incidence of ALS refers to newly diagnosed cases per year. The new text on page 2, lines 52-55, would be as follows:

"The global incidence of ALS is estimated at 1.75 cases per 100,000 inhabitants per year, with an incidence of 1.59 cases per 100,000 inhabitants per year in Europe and 1.4 cases per 100,000 inhabitants per year in Spain."

Comment 2: Page 9, line 247: This is a follow-up comment on Revision 22 – the p value (p = 2.17) appears to be a typo, as p values are between 0 and 1 by definition.

Response 2: Thank you for your observation. We have reviewed the reported p-value and confirmed that it is indeed a typographical error. As you correctly pointed out, p-values must be between 0 and 1.

The correct p-value for this test is “p= 0.17”, and we have amended the text in the manuscript on page 9, line 248 accordingly.

We appreciate your keen attention to detail, as this correction enhances the accuracy and rigor of the statistical analysis presented.

Comment 3: Page 10, Figure 2: This is a follow-up comment on Revision 25 – the authors’ explanation clarified that the direction and values of the Y axis cannot be standardized across figures since they reflect the inherent properties of different OPLS-DA models. However, it is still unclear what the values of Y axis represent (i.e., which specific parameters of the OPLS-DA model) in the bar plots. Also, panel b) seems to be a zoomed-in version of panel a), both representing the same OPLS-DA model, but these two panels are very different in the range of Y axis.

Response 3: Thank you for your insightful comment regarding the Y-axis representation in the OPLS-DA bar plots in Figure 2. We acknowledge that the description could have been clearer, and we appreciate the opportunity to improve the explanation.

The Y-axis in these bar plots represents the regression coefficients (weights) obtained from the OPLS-DA model. These coefficients indicate the contribution of each vocal parameter to the separation between groups (i.e., ALS subtypes vs. controls). Higher absolute values suggest a stronger influence of a given parameter on the discrimination model. Since OPLS-DA is a supervised multivariate technique, these weights result from a transformation of the original vocal measurements into a latent space that optimally separates the groups based on variance decomposition. We have now included this new section in the manuscript, in pages 10-11 and lines 294- 304:

“In the OPLS-DA bar plots, the Y-axis represents the regression coefficients (weights) assigned to each vocal parameter by the model. These coefficients indicate the relative contribution of each parameter to the discrimination between groups (e.g., ALS subtypes vs. controls). Higher absolute values suggest a stronger influence of a given parameter on the separation, with positive and negative values reflecting differences in feature expression between the compared groups. Since OPLS-DA is a supervised multivariate method, the original vocal parameters are transformed into a latent space that maximizes inter-group variance while minimizing intra-group variance. The regression coefficients in the bar plots result from this transformation and provide insight into which features most effectively distinguish ALS subtypes from controls. Importantly, the scale of the Y-axis is model-dependent and varies according to the magnitude of the coefficients derived from each specific comparison.”

Regarding panel (b), you are correct that it was intended as a zoomed-in version of panel (a) to highlight the parameters with the highest explanatory power. However, we acknowledge an error in the scale of the Y-axis, which made the two panels appear inconsistent. This has now been corrected in the revised figure to ensure proper alignment between the zoomed region and the full-scale plot. This is the new version of the figure:

We appreciate your careful review and constructive feedback, which have helped us refine both the figure and its explanation.

Comment 4: Pages 12-13, section 3.3 and Figure 5: This is a follow-up comment on Revision 28 – thank you for providing the detailed explanation in the cover letter. It could be helpful to add in the text that even though Figure 5 plotted the comparison between ALS and control groups, the model reported in this section focused on the Barthel Index within the ALS group and did not include ALS vs. control classification as an outcome variable. This might be applicable to Figure 6 as well.

Response 4: Thank you for your follow-up comment. We appreciate your suggestion for further clarification regarding the focus of the model in Section 3.3 and Figures 5 and 6.

To address this, we have now explicitly clarified in the text that, while Figure 5 presents a comparison between ALS and control groups, the OPLS-DA model described in this section is centered on the Barthel Index within the ALS group and does not include ALS vs. control classification as an outcome variable. Similarly, this clarification has also been incorporated into the description of Figure 6.

The following modifications have been made to the text:

On page 12, lines 363- 367, before describing the results of the OPLS-DA model, we have added: "It is important to note that, while Figure 5 visually presents the comparison between ALS and control groups, the OPLS-DA model reported in this section focuses exclusively on the Barthel Index within the ALS group. The classification between ALS and control individuals was not included as an outcome variable in this analysis."

On page 14, lines 388- 391 in the description of Figure 6, we have added: "As with the analysis presented in Figure 5, the OPLS-DA model used for Figure 6 was designed to assess the relationship between vocal parameters and ALSFRS-R scores within the ALS group, rather than to classify ALS versus control individuals."

We appreciate your insightful feedback, which has helped improve the clarity and interpretability of our manuscript.

Comment 5: Page 13, Figure 6: This is a follow-up comment on Revision 30. In lines 375-376, it states that the direction of the bars corresponds to whether the values were higher in the ALS or the control group. However, in line 378, it states that the direction reflects the associations between the ALSFRS-R scores and the vocal parameters (and based on the cover letter, it appeared that this association was only evaluated within the ALS group). It is still not very clear whether the direction of the bars represent the comparison of the values between ALS vs. control groups, or the correlations between ALSFRS-R and voice parameters within the ALS group. If the figure is based on the model assessing ALSFRS-R scores only in the ALS group, please consider renaming the arrows on the left side of the figure as “Positively/negatively correlated with ALSFRS-R scores”. This might be applicable to Figure 5 as well.

Response 5: Thank you for your follow-up comment. We appreciate the opportunity to clarify the meaning of the bar direction in Figure 6 and Figure 5.

To address this, we confirm that the Y-axis does not indicate positive or negative correlations. Instead, the values on the Y-axis represent the measured vocal parameters in ALS patients or in control individuals, depending on whether they are in the positive or negative range of the Y-axis.

To ensure this distinction is clear, we have made the following modifications:

Clarification in the text (page 14, lines 396-399), We have revised the explanation to explicitly state:

"The direction of the bars in Figure 6 does not indicate a positive or negative correlation but rather represents the measured vocal parameter values. Parameters plotted in the positive Y-axis region correspond to ALS patients, while those plotted in the negative Y-axis region correspond to controls as indicated."

Furthermore, to improve clarity, we have adjusted the labeling of the arrows on the left side of Figure 6 to: "Vocal parameter values in ALS” (in positive Y-axis) and “ Vocal parameter values in controls” (in negative Y-axis).

The same adjustment has been applied to Figure 5 for consistency.

These changes ensure that the figure legend and text accurately describe the meaning of the bar directions, aligning with the statistical model used.

We appreciate your keen attention to detail, which has helped refine the clarity and accuracy of our manuscript.

Reviewer 2 Report

Comments and Suggestions for Authors

I think that the comments have been adequately addressed. 

Author Response

Dear reviewer, thank you very much for your time and comments, which have undoubtedly contributed to improving the quality of our manuscript.

We are pleased to know that you feel we have been able to address them appropriately.

Round 3

Reviewer 1 Report

Comments and Suggestions for Authors

Thank you for the detailed revision and explanation. All of my comments on the previous version have been sufficiently addressed, except for just some minor follow-up comments to Response 5: Even though both figures are described as illustrating the differences in vocal parameters between ALS and control groups, the scales of the Y axis are very different (-0.025 to 0.015, vs. -8 to 12). Also, the directionality of some parameters is different between the two figures (e.g., Pr 18 is plotted a higher in control in Figure 5 but higher in ALS in Figure 6). If I understood correctly, the directionality of each parameter is based on whether the parameters were higher in ALS or control group in both figures, so that the directionality should stay the same. It could be helpful to double check the data used for plotting and briefly explain the inconsistencies between the two figures. It would also be helpful to clarify whether the plotted values are the measure values of individual groups (either ALS or control, as suggested in Response 5), or the differences between the two groups (as suggested in lines 369-370).

Author Response

Response to Reviewer Comment

Dear reviewer,

We thank you for the time and efforts you have put into reviewing our manuscript. We are now addressing your comments point by point, which will undoubtedly improve the quality of our paper. We take this opportunity to disclose the use of Al-based tools; these were used to assist editing under close supervision. We recognise that we should have included a statement about AI use on previous revisions. We note that, as stated in MDPI's guidelines, the use of Al tools for language enhancement is permitted when appropriately acknowledged (https://www.mdpi.com/ethics#_bookmark96).

Accordingly, we have now included a formal AI usage statement in the Acknowledgments section of the revised manuscript.

Regards

Comment Reviewer: Thank you for the detailed revision and explanation. All of my comments on the previous version have been sufficiently addressed, except for just some minor follow-up comments to Response 5: Even though both figures are described as illustrating the differences in vocal parameters between ALS and control groups, the scales of the Y axis are very different (-0.025 to 0.015, vs. -8 to 12). Also, the directionality of some parameters is different between the two figures (e.g., Pr 18 is plotted a higher in control in Figure 5 but higher in ALS in Figure 6). If I understood correctly, the directionality of each parameter is based on whether the parameters were higher in ALS or control group in both figures, so that the directionality should stay the same. It could be helpful to double check the data used for plotting and briefly explain the inconsistencies between the two figures. It would also be helpful to clarify whether the plotted values are the measure values of individual groups (either ALS or control, as suggested in Response 5), or the differences between the two groups (as suggested in lines 369-370).

Response: 1. Differences in Y-Axis Scaling:  

The Y-axis in Figure 5 represents the contribution of each vocal parameter to the OPLS-DA model based on the Barthel Index, while Figure 6 represents a different OPLS-DA model using ALSFRS-R as the predictor. Since these figures are derived from separate models with different response variables, their Y-axis scales are not directly comparable. This does not affect the interpretation, as the focus remains on the relative positioning of the parameters within each model.

In order to prevent any potential misunderstanding, we have now added this sentence at just before figure 6 (Page 15, lines 403- 407):

“It is important to note that Figure 5 and Figure 6 are based on separate OPLS-DA models, with Figure 5 using the Barthel Index as the response variable and Figure 6 using ALSFRS-R. As a result, their Y-axis scales are not directly comparable. However, this does not affect the interpretation, as the key focus remains on the relative positioning of the parameters within each model.”

  1. Directionality of Parameters Across Figures:

The directionality of parameters differs between Figures 5 and 6 because each figure is based on a distinct OPLS-DA model with a different clinical index. Figure 5 examines vocal parameter variations along the Barthel Index, while Figure 6 examines variations along the ALSFRS-R scale. Since these indices measure different aspects of ALS progression, certain parameters may have different relationships depending on which index is used as the response variable.

Importantly, parameters that are colored in blue—such as Pr18—have very low R2Y values, indicating that their contribution to the model is minimal and not statistically meaningful. As a result, their directionality should not be interpreted, as it does not carry relevant information about group differences. The key parameters to focus on are those highlighted in warm colors, which indicate stronger and more significant relationships with the respective clinical index.

We have now included a new section in M&M explaining these details (Page 8, lines 214-22):

“Furthermore, model loadings plots were color-coded, where features most strongly associated with the discriminant component were shown in warm colors (closer to red), while those in cold colors (closer to blue) were not considered discriminant. Therefore, parameters in blue represent features with very low R2Y values, indicating minimal contribution to the model and a lack of statistical significance. As such, their directionality or potential biological relevance should not be interpreted, as they do not provide meaningful insights into group differences. Instead, the primary focus should be on parameters highlighted in warm colors, which exhibit stronger and more significant associations with the respective clinical index”.

  1. Interpretation of Bar Values:

The bars in Figure 5 indicate the weighted contribution of each vocal parameter in differentiating individuals along the Barthel Index spectrum, whereas the bars in Figure 6 reflect contributions relative to ALSFRS-R. 

As example, in figure 5 positive values in the Y index indicate a stronger association with worse Barthel Index results (the ALS group), while negative values in the Y index indicate a stronger association with a better outcome in the evaluation (a better condition associated with the controls). This is different from raw mean comparisons and is specific to the predictive model used in each case.

We have now added a new section before figure 5 (page 15, lines 370- 385), the text now reads as follows:

“In the results of the O-PLS DA model using the Barthel Index, we observed that, regardless of sex or smoking status, ALS patients exhibited a significant relationship with Pr15 (degree of blockages in the total voice sample. A weaker association was also found with Pr14 (instability or inability to maintain amplitude across the entire voice sample) in individuals with ALS (Figure 5).

It is important to clarify that the Barthel Index is a numeric variable where 0 implies total dependence and 100 totally independent, and the OPLS-DA model in this analysis was designed to capture the relationship of this numeric variable with vocal parameters. The classification between ALS and control individuals was not included as an outcome variable in this model. However, for ease of the figure interpretation this includes labels indicating "ALS and control" based on known distributions of the Barthel Index (i.e. higher values associated with independence, and lower values associated with dependence).

These labels do not imply that a categorical classification was used in the model but rather help contextualize how vocal parameter variations align with different levels of functional impairment."

On the other hand, to avoid misunderstandings, we have eliminated the term “higher” in the information of the figure 5 Page 15 line 394 and 396, where it really means only “significant”.

We hope this explanation resolves the remaining concerns and appreciate the reviewer’s attention to detail. 
